# Towards an Understanding of Decision-Time vs. Background Planning in Model-Based Reinforcement Learning

## Abstract

In model-based reinforcement learning, an agent can leverage a learned model to improve its way of behaving in different ways. Two of the prevalent approaches are decision-time planning and background planning. In this study, we are interested in *understanding* under what conditions and in which settings one of these two planning styles will perform better than the other. After viewing them in a unified way through the lens of dynamic programming, we first consider the simplest instantiations of these planning styles and provide theoretical results and hypotheses on which one will perform better in the planning & learning and transfer learning settings. We then consider the modern instantiations of them and provide theoretical results and hypotheses on which one will perform better in the considered settings. Lastly, we perform several experiments to illustrate and validate both our theoretical results and hypotheses. Overall, our findings suggest that even though decision-time planning does not perform as well as background planning in its simplest instantiations, the modern instantiations of it can perform on par or better than the modern instantiations of background planning in both the planning & learning and transfer learning settings.

## 1 Introduction

It has long been argued that, in order for reinforcement learning (RL) agents to adapt to a variety of changing tasks, they should be able to learn a model of their environment, which allows for counterfactual reasoning and fast re-planning (Russell & Norvig, 2002). Although this is a widely-accepted view in the RL community, the question of *how* to leverage a learned model to perform planning does not have a widely-accepted and clear answer. In model-based RL, the two prevalent planning styles are decision-time and background planning (Sutton & Barto, 2018), where the agent mainly plans in the moment and in parallel to its interaction with the environment, respectively. In the context of RL, even though these two planning styles have been developed with different scenarios and application domains in mind:

- decision-time planning algorithms (Tesauro, 1994; Tesauro & Galperin, 1996; Silver et al., 2017; 2018) for scenarios in which the exact model of the environment is known and for domains that allow for an adequate computational budget at decision time (such as board games like chess and Go);

- background planning algorithms (Sutton, 1990; 1991; Łukasz Kaiser et al., 2020; Hafner et al., 2021; 2023) for scenarios in which the exact model is to be learned through pure interaction with the environment and for domains that are agnostic to the response time of the agent (such as gridworlds and arcade video games),

recently, with the introduction of the ability to learn a model through pure interaction (Schrittwieser et al., 2020), decision-time planning algorithms have been applied to the same scenarios and domains as their background planning counterparts (see e.g., Schrittwieser et al. (2020) and Hamrick et al. (2021) which both evaluate a decision-time planning algorithm, called MuZero, on Atari 2600 games). However, it still remains unclear under *what* conditions and in *which* settings one of these planning styles will perform better than the other.

In this study, we attempt to provide an answer to one aspect of this question. Specifically, we are interested in answering the following question:

> *Using the discounted return as the performance measure, under what conditions and in which settings will one planning style perform better than the other?*

To answer this question, we first start by abstracting away from the specific algorithmic details of the two planning styles and view them in a unified way through the lens of dynamic programming. Then, we consider the simplest instantiations of these planning styles and based on their dynamic programming interpretations and implementation details, provide theoretical results and hypotheses on which one will perform better in the planning & learning and transfer learning settings. We then consider the modern instantiations of these two planning styles and based on their dynamic programming interpretations and implementation details, provide theoretical results and hypotheses on which one will perform better in the considered settings. Lastly, we perform experiments with both instantiations to illustrate and validate our theoretical results and hypotheses .

Overall, our results suggest that even though decision-time planning does not perform as well as background planning in its simplest instantiations , due to (i) the improvements in the way planning is performed, (ii) the use of only real experience in the updates of the value estimates, and (iii) the ability to improve upon the existing policy at test time, the modern instantiations of it can perform on par or better than their background planning counterparts in both the planning & learning and transfer learning settings. We hope that our findings will help the RL community towards developing a better understanding of how the two planning styles compare against each other and stimulate research in improving them in potentially interesting ways.

## 2  Background

**Reinforcement Learning.** In RL (Sutton & Barto, 2018), an agent $A$ interacts with its environment $E$ through a sequence of actions to maximize its long-term cumulative reward. Here, the environment is usually modeled as a Markov decision process $E = (\mathcal{S}_E, \mathcal{A}_E, p_E, r_E, d_E, \gamma)$, where $\mathcal{S}_E$ and $\mathcal{A}_E$ are the (finite) set of states and actions, $p_E : \mathcal{S}_E \times \mathcal{A}_E \times \mathcal{S}_E \to [0,1]$ is the transition distribution, $r_E : \mathcal{S}_E \times \mathcal{A}_E \times \mathcal{S}_E \to \mathbb{R}$ is the reward function, $d_E : \mathcal{S}_E \to [0,1]$ is the initial state distribution, and $\gamma \in [0,1)$ is the discount factor. At each time step $t$, after taking an action $a_t \in \mathcal{A}_E$, the environment's state transitions from $s_t \in \mathcal{S}_E$ to $s_{t+1} \in \mathcal{S}_E$, and the agent receives an observation $o_{t+1} \in \mathcal{O}_E$ and an immediate reward $r_t$. As there is usually no prior access to the states in $\mathcal{S}_E$ and as the observations in $\mathcal{O}_E$ are usually high-dimensional, the agent has to operate on its own state space $\mathcal{S}_A$, which is generated by its own encoder $\phi : \mathcal{O}_E \to \mathcal{S}_A$. The goal of the agent is to jointly learn an encoder $\phi$ and a value estimator $Q : \mathcal{S}_A \to \mathbb{R}^{|\mathcal{A}_E|}$ that induces a policy $\pi \in \mathbb{\Pi} \equiv \{\pi | \pi : \mathcal{S}_A \times \mathcal{A}_E \to [0,1]\}$, maximizing $E_{\pi, p_E}[\sum_{t=0}^{\infty} \gamma^t r_E(S_t, A_t, S_{t+1})|S_0 \sim d_E]$.

**Model-Based RL.** One of the main ways of achieving this goal is through the use of model-based RL methods. In model-based RL, there are two alternating phases[1]: the learning and planning phases.[2] In the learning phase, the gathered experience is mainly used in jointly learning an encoder $\phi$ and a model $m \in \mathcal{M} \equiv \{(p_M, r_M, d_M)|p_M : \mathcal{S}_A \times \mathcal{A}_E \times \mathcal{S}_A \to [0,1], r_M : \mathcal{S}_A \times \mathcal{A}_E \times \mathcal{S}_A \to \mathbb{R}, d_M : \mathcal{S}_A \to [0,1]\}$, and optionally, the experience may also be used in jointly improving $\phi$ and $Q$.[3] In the planning phase, the learned model $m$ is then used for simulating experience, either to be used alongside real experience in improving the value estimator or just to be used in selecting actions at decision time. Also note that in the model-based RL literature it is usually implicitly assumed that $\mathcal{S}_E \subseteq \mathcal{S}_A$, which implies that the agent's model is capable of modeling what is going on underneath the environment.

---

[1] Note that even though some model-based algorithms, such as Ha & Schmidhuber (2018), first learn a model offline and then use it for planning in the rest of the agent-environment interaction, in this study, we will consider the default scenario in model-based RL where the agent constantly updates its model while interacting with the environment.

[2] Note that even though some model-based RL algorithms, such as Tesauro & Galperin (1996); Silver et al. (2017; 2018), do not employ a model learning phase and make use of an a priori given exact model, in this study, we will study versions of them in which the model has to be learned from pure interaction with the environment.

[3] Note that the learned model can both be in a parametric or non-parametric form (see van Hasselt et al. (2019)).

**Planning Styles in Model-Based RL.** In model-based RL, the two prevalent planning styles are decision-time and background planning (see Ch. 8 of Sutton & Barto (2018)).[4] Decision-time planning is performed as a computation whose output is the selection of a single action for the current state. This is often done by unrolling the model forward from the current state to compute local value estimates, which are then usually discarded after action selection. Here, planning is performed independently for *every* encountered state and it is mainly performed in an *online* fashion, though it may also contain offline components.

In contrast, background is performed by continually improving a cached value estimator, on the basis of simulated experience from the model, often in a global manner. Action selection is then quickly done by querying the value estimator at the current state. Unlike decision-time planning, background planning is often performed in a purely *offline* fashion, in parallel to the agent-environment interaction, and thus is *not* necessarily focused on the current state: well before action selection for any state, planning plays its part in improving the value estimates in many other states.

For convenience, in this study, we will refer to all model-based RL algorithms that have an online planning component as decision-time planning algorithms (see e.g., Tesauro (1994); Tesauro & Galperin (1996); Silver et al. (2017; 2018); Schrittwieser et al. (2020); Zhao et al. (2021)), and will refer to the rest as background planning algorithms (see e.g., Sutton (1990; 1991); Łukasz Kaiser et al. (2020); Hafner et al. (2021; 2023); Zhao et al. (2021)). Note that, regardless of the style, any type of planning can be viewed as a procedure $f : (\mathcal{M}, \mathbb{\Pi}) \to \mathbb{\Pi}$, that takes a model $m$ and a policy $\pi^i$ as input and returns an improved policy $\pi_m^o$, according to $m$, as output.

**Categorization within the Two Planning Styles.** Starting with decision-time planning, depending on how much search is performed, decision-time planning algorithms can be studied under three main categories:

1. Decision-time planning algorithms that perform no search (see e.g., Tesauro & Galperin (1996) and Alg. 1)

2. Decision-time planning algorithms that perform pure search (see e.g., Campbell et al. (2002) and Alg. 2)

3. Decision-time planning algorithms that perform some amount of search (see e.g., MuZero (Schrittwieser et al., 2020) and Alg. 7)

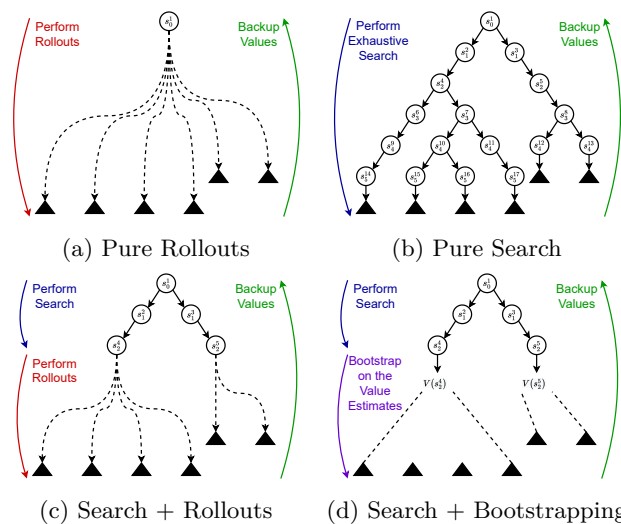

(a) Pure Rollouts      (b) Pure Search

(c) Search + Rollouts      (d) Search + Bootstrapping

Figure 1: The different planning styles within decision-time planning in which planning is performed (a) by purely performing rollouts, (b) by purely performing search, (c) by performing rollouts after performing some amount of search and (d) by bootstrapping on the value estimates after performing some amount of search. The subscripts and superscripts on the states indicate the time steps and state identifiers, respectively. The black triangles indicate the terminal states.

In the first two of these categories, planning is performed (i) by just running pure rollouts with a fixed / improving policy (see Fig. 1a), and (ii) by purely performing search (see Fig. 1b), respectively. In the last one, planning is performed by first performing some amount of search and then either (i) by running rollouts with a fixed / improving policy, (ii) by bootstrapping on the cached value estimates of a fixed / improving policy, or (iii) by doing both (see Fig. 1c & 1d). Note that while the simplest instantiations of decision-time planning fall within the first two categories, the modern instantiations of it fall within the last one. Also note that, while planning is performed with only a single parametric model in the first two categories, it is usually performed with both a parametric and

---

[4]Although some new planning styles have been proposed in the transfer learning literature (see e.g., Barreto et al. (2017; 2019; 2020); Alver & Precup (2022)), these approaches can also be viewed as performing some form of decision-time planning with pre-learned models.

non-parametric (usually a replay buffer, see van Hasselt et al. (2019)) model in the last one. We refer the reader to Bertsekas (2021) for more details on the different categories of decision-time planning algorithms.

Moving on to background planning, as all background planning algorithms (see e.g., Dyna Sutton (1990; 1991), Dreamer (Hafner et al., 2021; 2023) and Alg. 3, 4, 8) perform planning by continually improving a cached value estimator throughout the model learning process, we do not study them under different categories. However, we again note that while the simplest instantiations of background planning perform planning with a single parametric model (see e.g., Alg. 3 & 4), the modern instantiations of it perform planning with both a parametric and non-parametric (usually a replay buffer) model (see e.g., Alg. 8).

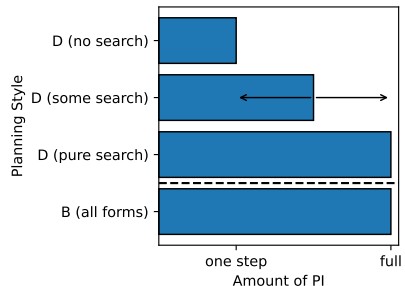

Figure 2: The amount of PI that decision-time (D) and background (B) planning corresponds to at each time step.

## 3 A Unified Dynamic Programming View of the Two Planning Styles

In this section, we abstract away from the algorithmic details, such as whether policy improvement is done locally or globally, or whether planning is performed in an online or offline manner, and view the two planning styles in a unified way through the lens of dynamic programming (Bertsekas & Tsitsiklis, 1996). More specifically, we view decision-time and background planning through the lens of the well-known policy iteration (PI) algorithm.[5] In this framework, the two planning styles can be viewed as follows:

- Decision-time planning algorithms that perform **no search** can be considered as performing one-step PI on top of a fixed / improving policy at every time step, as at each time step they compute a $\pi_m^o$ by first running many rollouts in $m$ with a fixed /improving $\pi^i$ to evaluate the current state (which corresponds to policy evaluation) and then selecting the most promising action (which corresponds to policy improvement).

- Similarly, decision-time planning algorithms that perform **pure search** can be considered as performing PI until convergence (which we call full PI) at every time step, as at each time step they disregard $\pi^i$ and compute a $\pi_m^o$ by first performing exhaustive search in $m$ to obtain the optimal values at the current state and then selecting the most promising action.

- Finally, decision-time planning algorithms that perform **some amount of search** can be considered as performing an amount of PI that is between one-step and full PI on top of a fixed / improving policy at every time step, as they are at the intersection of decision-time planning algorithms that perform no search and pure search.

- All background planning algorithms can be considered as performing an amount of PI that is usually less than one-step PI on top of an improving policy at every time step, as at each time step they compute a $\pi_m^o$ by gradually improving $\pi^i$ on the basis of simulated experience from $m$. However, if the learned model $m$ converges in the model learning process, **all** background planning algorithms can be considered as performing an amount of PI that is eventually equivalent to full PI, as the continual improvements to $\pi^i$ at each time step would eventually lead to an improvement that is equivalent to performing full PI.

See Fig. 2 for a graphical depiction of the amount of PI that each planning style corresponds to at each time step. Note that while the PI view of decision-time planning abstracts a planning process that focuses on the agent's current state, the PI view of background planning abstracts a one that is dispersed across the agent-environment interaction. Also note that, in the end of Sec. 2, we have pointed out that some decision-time

---

[5]We choose policy iteration over value iteration as it better describes how planning is performed in decision-time planning (see Bertsekas (2021)), and it is also useful in describing the planning process in background planning.

and background planning algorithms perform planning with both a parametric and non-parametric model, which can make it difficult for them to be viewed through the proposed unified framework. However, if one considers the two separate models as a single combined model, then these algorithms can also be viewed straightforwardly in our proposed framework. We refer the reader to App. A for a broader discussion.

## 4 Performance Measure and Partitioning of the Model Space

**Performance Measure.** In this study, we are interested in understanding under what conditions and in which settings will one planning style perform better than the other. Thus, we start by formally defining a performance measure that will be used in comparing the two planning styles of interest. Given an arbitrary model $m = (p, r, d) \in \mathcal{M}$, let us define the performance of an arbitrary policy $\pi \in \Pi$ in it as follows:

$$J_m^\pi \equiv E_{\pi,p}\left[\sum_{t=0}^\infty \gamma^t r(S_t, A_t, S_{t+1})\bigg| S_0 \sim d\right]. \tag{1}$$

Note that $J_m^\pi$ corresponds to the expected *discounted* return of a policy $\pi$ in model $m$. Next, we start considering the conditions under which the comparisons will be made: we are interested in both simple scenarios in which the value estimators and models are represented as a table, and in complex ones in which they are represented using function approximation.

**Partitioning of the Model Space.** Before moving on to the comparison between the two planning styles, we first present a way to partition the space of agent models $\mathcal{M}$ such that it would be possible to understand when will one planning style be guaranteed to perform on par or better than the other. Let us start by defining $m^*$ to be the exact model of the environment. Note that $m^* \in \mathcal{M}$ as $\mathcal{S}_E \subseteq \mathcal{S}_A$ (see Sec. 2). Then, given a policy set $\Pi \subseteq \Pi$ containing at least two policies and a performance measure $J$ defined as in (1), depending on the relative performances of the policies in it and in $m^*$, a model $m \in \mathcal{M}$ can belong to one of the following main classes:

**Definition 1** (PCM). *Given a $\Pi \subseteq \Pi$ and a $J$, let*

$$\mathcal{M}_{\Pi,J}^{\mathrm{PCM}} \equiv \{m \in \mathcal{M} \mid J_{m^*}^{\pi^i} \leq J_{m^*}^{\pi^j} \text{ for all } \pi^i, \pi^j \in \Pi \text{ satisfying } J_m^{\pi^i} \geq J_m^{\pi^j}$$
$$\text{and } J_{m^*}^{\pi^i} \geq J_{m^*}^{\pi^j} \text{ for all } \pi^i, \pi^j \in \Pi \text{ satisfying } J_m^{\pi^i} \leq J_m^{\pi^j}\}.$$

*We say that each $m \in \mathcal{M}_{\Pi,J}^{\mathrm{PCM}}$ is a* performance-contrasting model (PCM) *of $m^*$ with respect to $\Pi$ and $J$.*

**Definition 2** (PRM). *Given a $\Pi \subseteq \Pi$ and a $J$, let*

$$\mathcal{M}_{\Pi,J}^{\mathrm{PRM}} \equiv \{m \in \mathcal{M} \mid J_{m^*}^{\pi^i} \geq J_{m^*}^{\pi^j} \text{ for all } \pi^i, \pi^j \in \Pi \text{ satisfying } J_m^{\pi^i} \geq J_m^{\pi^j}$$
$$\text{and } J_{m^*}^{\pi^i} \leq J_{m^*}^{\pi^j} \text{ for all } \pi^i, \pi^j \in \Pi \text{ satisfying } J_m^{\pi^i} \leq J_m^{\pi^j}\}.$$

*We say that each $m \in \mathcal{M}_{\Pi,J}^{\mathrm{PRM}}$ is a* performance-resembling model (PRM) *of $m^*$ with respect to $\Pi$ and $J$.*

Informally, given any two policies in $\Pi$ and a $J$, (i) a model $m$ is a PCM of $m^*$ if the policy that performs on par or better in it performs on par or worse in $m^*$ and (ii) it is a PRM of $m^*$ if the policy that performs on par or better in it also performs on par or better in $m^*$.[6] If $\Pi$ contains at least one of the optimal policies for $m$, then $m$ can also belong to one of the following specialized classes:

**Definition 3** (PNM). *Given a $\Pi \subseteq \Pi$ and a $J$, let*

$$\mathcal{M}_{\Pi,J}^{\mathrm{PNM}} \equiv \{m \in \mathcal{M}_{\Pi,J}^{\mathrm{PCM}} \mid J_{m^*}^{\pi_m^*} = \min_{\pi \in \Pi} J_{m^*}^\pi \text{ for all } \pi_m^* \in \Pi\},$$

*where $\pi_m^*$ denotes the optimal policies in $m$. We say that each $m \in \mathcal{M}_{\Pi,J}^{\mathrm{PNM}}$ is a* performance-minimizing model (PNM) *of $m^*$ with respect to $\Pi$ and $J$.*

**Definition 4** (PXM). *Given a $\Pi \subseteq \Pi$ and a $J$, let*

$$\mathcal{M}_{\Pi,J}^{\mathrm{PXM}} = \{m \in \mathcal{M}_{\Pi,J}^{\mathrm{PRM}} \mid J_{m^*}^{\pi_m^*} = \max_{\pi \in \Pi} J_{m^*}^\pi \text{ for all } \pi_m^* \in \Pi\},$$

*where $\pi_m^*$ denotes the optimal policies in $m$. We say that each $m \in \mathcal{M}_{\Pi,J}^{\mathrm{PXM}}$ is a* performance-maximizing model (PXM) *of $m^*$ with respect to $\Pi$ and $J$.*

---

[6]Note that $m$ can both be a PCM and a PRM of $m^*$ if the two policies perform on par in both $m$ and $m^*$.

Informally, given a subset of $\Pi$ containing the optimal policies for model $m$ and a $J$, (i) $m$ is a PNM of $m^*$ if all of the optimal policies result in the worst possible performance in $m^*$ and (ii) it is a PXM of $m^*$ if all them result in the best possible performance in $m^*$. Note that (i) PNMs are a subclass of PCMs and (ii) PXMs are a subclass of PRMs. Also note that the definitions above are agnostic to how the models are represented, i.e., whether they are represented through tables or function approximators.

Fig. 3a illustrates how $\mathcal{M}$ is generally partitioned for an arbitrary $\Pi$ and $J$. Note that given a fixed $J$, the relative sizes of the model classes solely depend on $\Pi$. For instance, as $\Pi$ gets larger, the relative sizes of $\mathcal{M}_{\Pi,J}^{\mathrm{PCM}}$ and $\mathcal{M}_{\Pi,J}^{\mathrm{PRM}}$ shrink, because with every policy that is added to $\Pi$, the number of criteria that a model must satisfy to be a PCM or PRM increases, which reduces the odds of an arbitrary model in $\mathcal{M}$ being in $\mathcal{M}_{\Pi,J}^{\mathrm{PCM}}$ or $\mathcal{M}_{\Pi,J}^{\mathrm{PRM}}$ (see the blue region in Fig. 3a). And, as $\Pi$ gets smaller, the relative sizes of $\mathcal{M}_{\Pi,J}^{\mathrm{PCM}}$ and $\mathcal{M}_{\Pi,J}^{\mathrm{PRM}}$ grow, and eventually fill up the entire space when $\Pi$ contains only two policies. Fig. 3b illustrates the partitioning in this scenario. Since we are only interested in comparing the policies of two planning styles, the $\Pi$ of interest has a size of two, i.e. $|\Pi| = 2$, and thus we have a partitioning as in Fig. 3b which covers the *entire* model space $\mathcal{M}$.

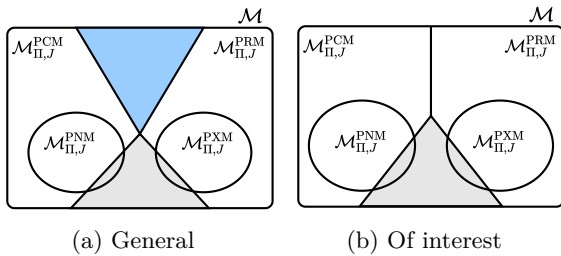

(a) General      (b) Of interest

Figure 3: (a) The general partitioning and (b) the partitioning of interest of $\mathcal{M}$, for a given $\Pi$ and $J$. The gray and blue regions indicate $\mathcal{M}_{\Pi,J}^{\mathrm{PCM}} \cap \mathcal{M}_{\Pi,J}^{\mathrm{PRM}}$ and $\mathcal{M} \setminus (\mathcal{M}_{\Pi,J}^{\mathrm{PCM}} \cup \mathcal{M}_{\Pi,J}^{\mathrm{PRM}})$, respectively.

**Illustrative Example.** As an illustration of the model classes defined above, let us start by considering the Simple Gridworld environment depicted in Fig. 4, in which the agent spawns in state S and has to navigate to the goal state G. At each time step, the agent receives an $(x, y)$ pair, indicating its position, and based on this selects an action that moves it to one of the four neighboring cells with a slip probability of 0.05. The agent receives a negative reward that is linearly proportional to its distance from G and a reward of $+10$ if it reaches G. In this environment, given a policy set $\Pi$ containing the policies of the two planning styles and the performance measure $J$, examples of PCMs and PRMs can be tabular models with goal states of $\{G_n\}_{n=2}^5$ and $\{G_n\}_{n=6}^9$, respectively. And, assuming that $\Pi$ contains at least one of the optimal policies, examples of a PNM and a PXM can be tabular models with goal states of $G_1$ and G, respectively.

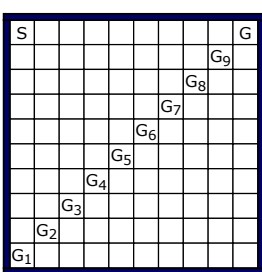

Figure 4: The Simple Gridworld environment.

Even though we have provided definitions for four different model classes, throughout the rest of this study, we will only provide theoretical results and hypotheses for the scenarios in which the models of the two planning styles converge to PRMs and PXMs, as the model learning process pushes their models towards becoming either PRMs or PXMs, even if they are initialized as PCMs or PNMs. We have only provided definitions for PCMs and PNMs to paint a complete picture of the space of possible models.

# 5 Decision-Time vs. Background Planning

## 5.1 Simplest Instantiations of the Two Planning Styles

We are now ready to discuss when will one planning style perform better than the other across different conditions and settings. For easy analysis, we start by considering the simplest instantiations of the two planning styles, which can be found in Ch. 8 of Sutton & Barto (2018). More specifically, for decision-time planning we study a version of the online Monte-Carlo planning algorithm of Tesauro & Galperin (1996) in which a parametric model is learned from experience (see Alg. 1) and for background planning we study a version the Dyna-Q algorithm of Sutton (1990; 1991) in which the value estimator is updated by using

samples from only the model (see Alg. 4).[7] We refer the reader to App. B for a discussion on why we consider these specific versions.

In our proposed framework (see Sec. 3), these algorithms can be viewed as follows:

- As the decision-time planning algorithm performs planning by first running many rollouts with a fixed policy in the model and then by selecting the most promising action, it can be considered as performing one-step PI on top of a fixed policy at every time step.

- And, as the background planning algorithm performs planning by continually improving a value estimator at every time step with samples from the model, it can be viewed as performing an amount of PI that is eventually equivalent full PI when its learned model converges.

Note that, although we only consider these specific instantiations, as long as decision-time planning corresponds to taking a smaller or on par policy improvement step than background planning, the theoretical results and hypotheses that we provide in this section would hold regardless of the choice of instantiation.

Before considering different conditions and settings, let us define the following policies that will be useful in referring to the input and output policies of the two planning styles:

**Definition 5** (Base, Rollout (Bertsekas, 2021) and Certainty-Equivalence (Jiang et al., 2015) Policies). *The* base policy $\pi^b \in \Pi$ *is the policy that is used in initiating PI. Given a base policy $\pi^b$ and a model $m \in \mathcal{M}$, the* rollout policy $\pi^r_m \in \Pi$ *is the policy obtained after performing one-step of PI on top of $\pi^b$ in $m$, and the* certainty-equivalence policy $\pi^{ce}_m \in \Pi$ *is the policy obtained after performing full PI in $m$.*

In the rest of this section, we will refer to the policies generated by the simplest instantiations of decision-time and background planning with model $m$ as $\pi^r_m$ and $\pi^{ce}_m$, respectively.

### 5.1.1 Planning & Learning Setting

**Fairest Scenario.** For the fairest possible comparison, we start by considering the scenario in which the two planning styles would perform planning with the same model $m \in \mathcal{M}$, that is to be learned in the model learning process. In this scenario, when the value estimators of both planning styles are represented as a table, we can prove the following statement:

**Proposition 1.** *Let $m \in \mathcal{M}$ be a PRM of $m^*$ with respect to $\Pi = \{\pi^r_m, \pi^{ce}_m\} \subseteq \Pi$ and $J$. Then, $J^{\pi^{ce}_m}_{m^*} \geq J^{\pi^r_m}_{m^*}$.*

Due to space constraints, we defer the proofs to App. C. Prop. 1 implies that, given $\Pi = \{\pi^r_m, \pi^{ce}_m\}$ and $J$, decision-time planning will perform on par or worse than background planning if $m$ converges to a PRM. Note that even though this result would not be guaranteed to hold if function approximation was to be used in the value estimator representations[8], if one were to use approximators with good generalization capabilities (i.e., approximators that assign the same value to similar observations), we would expect a similar performance trend to hold.

To put it more explicitly, in the fairest scenario, we would expect the following statements to hold:

> **Theoretical Result 1.** When the value estimators of both planning styles are represented as tables, decision-time planning will perform or par or worse than background planning if $m$ converges to a PRM.
>
> **Hypothesis 1.** When the value estimators of both planning styles are represented with function approximators that have good generalization capabilities, we would expect a similar performance trend with Theoretical Result 1.

---

[7]Note that for the simplest instantiations of decision-time planning, we choose to study an algorithm that performs no search, and not a one that performs pure search (see Sec. 2), as the latter ones require a significant amount of computational budget at decision-time and thus are not practically applicable to most scenarios.

[8]As in this case, there would be no guarantee that full PI will result in a better policy than one-step PI in $m$, which is a basic result from dynamic programming (Bertsekas & Tsitsiklis, 1996).

**Common Scenario.** In common comparison scenarios, instead of restricting the two planning styles to perform planning with the same model, the comparison is usually done by allowing the two planning styles to perform planning with their own models, that are again to be learned in the model learning process. In this scenario, as different trajectories are likely to be followed in the model learning process, the encountered models of the two planning styles, which we denote as $m_d \in \mathcal{M}$ and $m_b \in \mathcal{M}$ for decision-time and background planning, respectively, are also likely to be different. Thus, even though coming up with a theoretical result that is as strong as Prop. 1 is not possible, when the value estimators of both planning styles are represented as a table, we can still prove the following statement:

**Proposition 2.** *Let $m_d \in \mathcal{M}$ be any model of $m^*$ with respect to $\Pi_d = \{\pi^r_{m_d}, \pi^{ce}_{m_d}\} \subseteq \Pi$ and $J$, and let $m_b \in \mathcal{M}$ be a PXM of $m^*$ with respect to $\Pi_b = \{\pi^r_{m_b}, \pi^{ce}_{m_b}\} \subseteq \Pi$ and $J$. Then, $J^{\pi^{ce}_{m_b}}_{m^*} \geq J^{\pi^r_{m_d}}_{m^*}$.*

Prop. 2 implies that, given $\Pi_d = \{\pi^r_{m_d}, \pi^{ce}_{m_d}\}$, $\Pi_b = \{\pi^r_{m_d}, \pi^{ce}_{m_d}\}$ and $J$, decision-time planning will perform on par or worse than background planning if $m_b$ converges to a PXM. Note again that even though Prop. 2 would not be guaranteed to hold if function approximation was to be used in the value estimator representations[9], if one were to use approximators with good generalization capabilities, we would again expect a similar performance trend to hold.

More explicitly, in the common scenario, we would expect the following statements to hold:

> **Theoretical Result 2.** When the value estimators of both planning styles are represented as tables, decision-time planning will perform or par or worse than background planning if $m_b$ converges to a PXM.
>
> **Hypothesis 2.** When the value estimators of both planning styles are represented with function approximators that have good generalization capabilities, we would expect a similar performance trend with Theoretical Result 2.

### 5.1.2 Transfer Learning Setting

**Adaptation Scenario.** Although there are many different scenarios in the transfer learning setting (Taylor & Stone, 2009), for easy analysis, we start by considering a simple and commonly used one in which (ii) there is only one training task and a subsequent test task that differs only in the reward function and in which (ii) the agent's transfer ability is measured by how fast it adapts to the test task after being trained on the training task.[10] We refer to this transfer learning setting as the adaptation scenario. In this scenario, we would expect the statements of the common planning & learning scenario to hold directly, as instead of a single one, there are now two consecutive common planning & learning scenarios.

Restating more clearly, in the adaptation scenario, we would expect the following statements to hold:

> **Theoretical Result 3.** When the value estimators of both planning styles are represented as tables, decision-time planning will first perform or par or worse than background planning if $m_b$ converges to a PXM, and the same would happen in the subsequent test task.
>
> **Hypothesis 3.** When the value estimators of both planning styles are represented with function approximators that have good generalization capabilities, we would expect a similar performance trend with Theoretical Result 3.

### 5.2 Modern Instantiations of the Two Planning Styles

We now consider the modern instantiations of the two planning styles. More specifically, for decision-time planning we study both the decision-time planning algorithm in Zhao et al. (2021) (see Alg. 7) and MuZero

---

[9]Due to the same reason discussed in the fairest comparison scenario.
[10]More challenging settings will also be considered in the next section.

(Schrittwieser et al., 2020). And, for background planning, we study both the background planning algorithm in Zhao et al. (2021) (see Alg. 8) and DreamerV3 (Hafner et al., 2023).[11]

In our proposed framework (see Sec. 3), these algorithms can be viewed as follows:

- As the decision-time planning algorithms perform planning by first performing some amount of search and then by bootstrapping on the value estimates of a continually improving policy, they can be considered as performing more than one-step but less than full PI on top of an improving policy at every time step.

- And, as the background planning algorithms perform planning by continually improving a value estimator at every time step with samples from the model, they can be viewed as performing an amount of PI that is eventually equivalent to full PI when its learned model converges.

Note that although we only consider these specific instantiations, the theoretical results and hypotheses we provide in this section are also generally applicable to most state-of-the-art model-based RL algorithms (Moerland et al., 2023), as these istantiations are reflective of many of their properties.

### 5.2.1 Planning & Learning Setting

**Simplified Scenario.** To ease the analysis, we start by considering a simplified scenario in which both the value estimators and models of the modern instantiations are represented as a table. Let us also define the *improved rollout policy* to be as follows:

**Definition 6** (Improved Rollout Policy). *Given a base policy $\pi^b$ and a model $m \in \mathcal{M}$, the* improved rollout policy $\pi_m^{r+} \in \overline{\Pi}$ *is the policy obtained after performing more than one-step but less than full PI on top of $\pi^b$ in $m$.*

And, let us also refer to the policies generated by the modern instantiations of decision-time and background planning with models $m_d$ and $m_b$ as $\pi_{m_d}^{r+}$ and $\pi_{m_b}^{ce}$, respectively. Then, using $\pi_{m_d}^{r+}$ and $\pi_{m_b}^{ce}$ in place of $\pi_{m_d}^{r}$ and $\pi_{m_b}^{ce}$, respectively, we would expect Prop. 2 to hold exactly as decision-time planning still corresponds taking a smaller or on par policy improvement step than background planning. However, as decision-time planning now corresponds to performing more than one-step PI, we would expect the performance gap between the two planning styles to reduce in their modern instantiations . Moreover, we would expect this gap to gradually close if both $m_d$ and $m_b$ converge to PXMs, as the use of an improving base policy for decision-time planning would result in a continually improving performance that gets closer to the one of background planning.

More explicitly, in the simplified scenario, we would expect the following statement to hold:

> **Theoretical Result 4.** When the value estimators of both planning styles are represented as tables, decision-time planning will now catch up with the performance background planning if both $m_d$ and $m_b$ converge to PXMs.

**Original Scenario.** We now consider the original scenario in which both the value estimators and models of the two planning styles are represented with neural networks. In this scenario, we would expect a similar performance trend to hold as neural networks are approximators with good generalization capabilities. However, note that this expectation is solely based on our abstract dynamic programming view and thus does not take into consideration the issues that may arise in practice when background planning is implemented with neural networks, which can also play an important role on how the two planning styles will compare against each other.

More specifically, when neural networks are used in the representation of the model, unlike the simplified scenario, the model is likely to hallucinate observations (or states) that do not actually exist in the original environment, which is usually known as "hallucinated observations" (Jafferjee et al., 2020). And, as background

---

[11]We choose to study the algorithms in Zhao et al. (2021) in addition to the state-of-the-art algorithms MuZero and DreamerV3 as they are *generic* algorithms that are reflective of many of the properties of their state-of-the-art counterparts.

planning performs planning by updating its value estimator with the simulated experience that is generated by its model, it is performs updates to its value estimator with these "hallucinated observations" (see line 20 in Alg. 8), which can prevent it from reaching optimal or good performance (see e.g., van Hasselt et al. (2019); Jafferjee et al. (2020)). Note that this is not a problem in decision-time planning as it performs updates to its value estimator with only the real experience. Thus, we hypothesize that compared to decision-time planning, it is likely for background planning to suffer more in reaching optimal or good performance when their models are implemented with neural networks.

Putting it more explicitly, in the original scenario, we would expect the following statement to hold:

> **Hypothesis 4.** When the models of both planning styles are represented with neural networks, we would expect deviations from Theoretical Result 4 and decision-time planning would perform better than background planning.

### 5.2.2 Transfer Learning Setting

**Adaptation and Zero-shot Scenarios.** We now consider two common scenarios that are both more challenging than the setting considered in Sec. 5.1. In these scenarios, there is a distribution of training and test tasks, differing only in their observations. In the first one, the agent's transfer ability is measured by how fast it adapts to the test tasks after being trained on the training tasks, which we again refer to as the adaptation scenario (see e.g., Van Seijen et al. (2020)), and in the second one, this ability is measured by the agent's instantaneous performance on the test tasks as it gets trained on the training tasks, which we refer to as the zero-shot scenario (see e.g., Zhao et al. (2021); Anand et al. (2022)). Note that, in these scenarios, as there is a distribution of tasks, the use of neural networks in both the value estimator and model of the two planning styles is inevitable.

In both scnearios we would again expect "hallucinated observations" to prevent background planning in reaching optimal or good performance on the training tasks because of the same reasons discussed in the planning & learning setting (Sec. 5.2.1). Additionally, in the adaptation scenario, after the tasks switch from the training tasks to the test tasks, we would expect background planning to suffer more in the adaptation process, as its model would keep "hallucinating" experience that resembles the training tasks until it adapts to the test tasks, which in the meantime would lead to harmful updates to its value estimator. Also, in the zero-shot scenario, if the model of decision-time planning becomes capable of simulating at least a few time steps of the test tasks, we would expect decision-time planning to perform better on the test tasks, as at test time it would be able to improve upon its existing policy by performing online planning.[12]

More concretely, in the adaptation and zero-shot scenarios, we would expect the following statements to hold:

> **Hypothesis 5.** In both the adaptation and zero-shot scenarios, we would expect a similar performance trend with Hypothesis 4 on the training tasks.
>
> **Hypothesis 6.** In the adaptation scenario, we would expect background planning to suffer more in the adaptation process and perform worse than decision-time planning on the test tasks.
>
> **Hypothesis 7.** In the zero-shot scenario, if the model of decision-time planning becomes capable of simulating at least a few time steps of the test tasks, we would expect decision-time planning to improve upon its existing policy and perform better than background planning on the test tasks.

---

[12]Note that, it is usually the case that the model of decision-time planning becomes capable of simulating at least a few time steps of the test tasks. Also note that improving upon its existing policy is not possible for background planning, as it performs planning in an offline fashion and thus requires additional interaction with the test tasks (which is not possible in the zero-shot setting).

# 6 Experiments and Results

We now perform experiments to illustrate and validate the theoretical results and hypotheses presented in Sec. 5. The experimental details can be found in App. D.

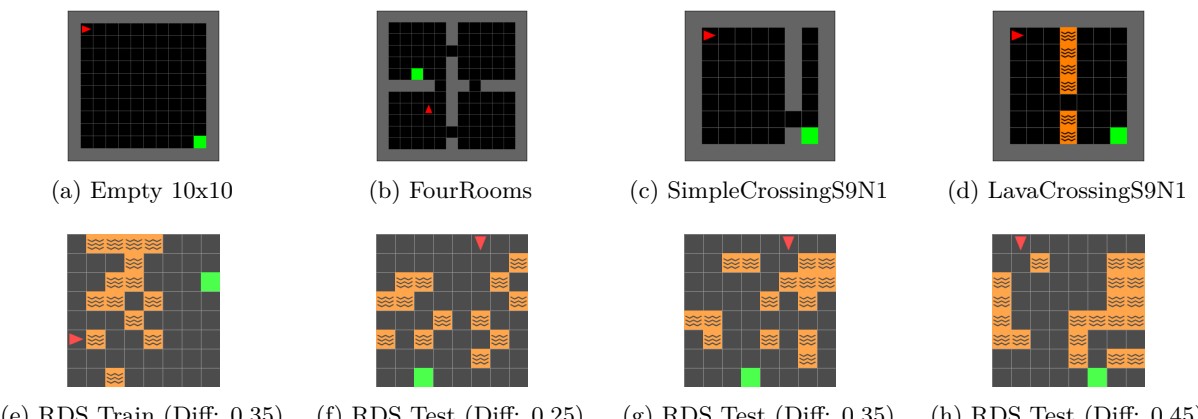

(a) Empty 10x10    (b) FourRooms    (c) SimpleCrossingS9N1    (d) LavaCrossingS9N1

(e) RDS Train (Diff: 0.35)    (f) RDS Test (Diff: 0.25)    (g) RDS Test (Diff: 0.35)    (h) RDS Test (Diff: 0.45)

Figure 5: (a-d) The Empty 10x10, FourRooms, SimpleCrossingS9N1 and LavaCrossingS9N1 environments in MiniGrid. (e-h) The training task of difficulty 0.35 and test tasks of difficulties 0.25, 0.35 and 0.45 in the RDS environment (Zhao et al., 2021). Note that the difficulty parameter here controls the density of the lava cells between the agent and the goal cell, and that the test tasks are just transposed versions of the training tasks. Also note that with every reset of the episode, a new lava cell pattern is procedurally generated for both the training and test tasks. More on the details of the RDS environment can be found in Zhao et al. (2021).

**Environmental Details.** As a testbed, we use four different domains: (i) the Simple Gridworld environment (see Fig. 4), (ii) five environments from MiniGrid (see Fig. 5, Chevalier-Boisvert et al., 2018), (iii) four environments from the Atari suite (Bellemare et al., 2013), and (iv) four environemnts from the Procgen benchmark (Cobbe et al., 2020). We choose the former two domains as the optimal policies in them are easy to learn and thus they allow for designing controlled experiments that are helpful in answering the questions of interest to this study. We choose the latter two as they allow for demonstrating the scalability of our statements in more complex scenarios.

The details of the Simple Gridworld environment are already presented in Sec. 4. In MiniGrid environments, the agent (depicted in red) has to navigate to the green goal cell, while avoiding the orange lava cells (if there are any). At each time step, the agent receives a top-down image of the grid and based on this chooses an action that moves it to one of the four neighboring cells. If the agent steps on a lava cell, the episode terminates with no reward, and if it reaches the goal cell, the episode terminates with a reward of $+1$. More details on Minigrid environments can be found in App. D. And, for the details of the Atari and Procgen environments, we refer the reader to studies that introduced them. Note that while $\mathcal{O}_E = \mathcal{S}_E$ in the Simple Gridworld environment, $\mathcal{O}_E \neq \mathcal{S}_E$ in the MiniGrid, Atari and Procgen environments.

## 6.1 Experiments with Simplest Instantiations

In this section, we perform experiments with the simplest instantiations of decision-time and background planning (see Alg. 1 & 4) on the Simple Gridworld environment to illustrate and validate our theoretical results and hypotheses presented in Sec. 5.1. In addition to the scenario in which the value estimators are represented as tables, we also consider a one in which we use state aggregation in the value estimator representation, i.e., $\phi$ is a state aggregator (see Fig. D.1d). More on the implementation details of these instantiations can be found in App. D.1.

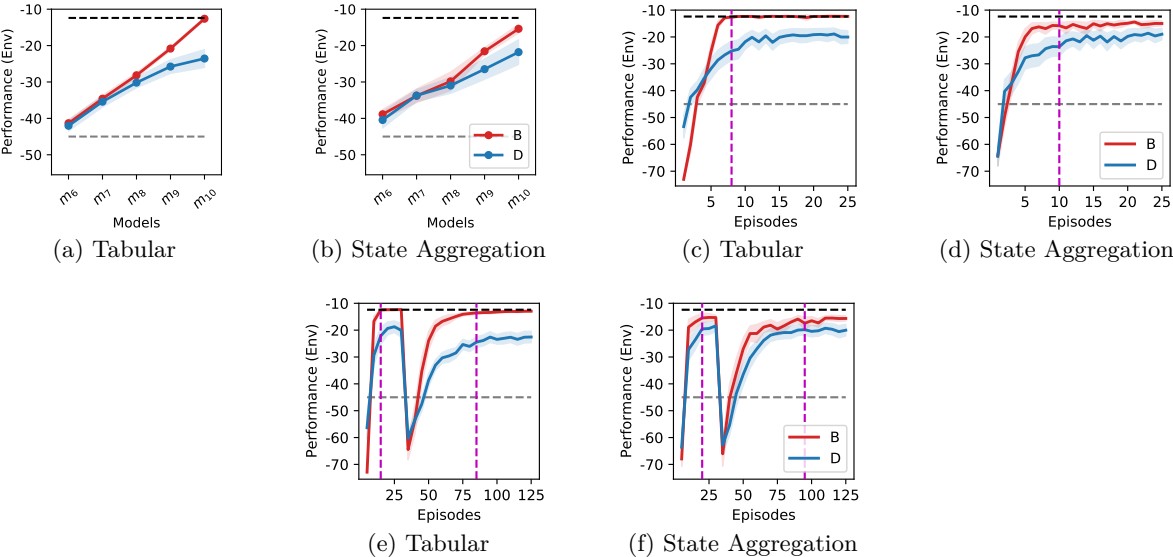

Figure 6: The performance of the simplest instantiations of decision-time (D) and background (B) planning on the Simple Gridworld environment, in the (a, b, c, d) planning & learning and (e, f) transfer learning settings with tabular and state aggregation value estimator representations. Black & gray dashed lines indicate the performance of the optimal & random policies, respectively. The magenta dashed line in (c, d, e, f) indicates the point after which background planning's model becomes and remains as a PXM. Shaded regions are one standard error over 250 runs.

### 6.1.1 Planning & Learning Experiments

**Fairest Scenario.** According to Theoretical Result 1, when tabular value estimators are used in the simplest instantiations, decision-time planning is guaranteed to perform on par or worse than background planning if their model $m$ converges to a PRM. For empirical illustration, we designed a controlled setting in which we trained $m$ to sequentially converge to a set of PRMs: $m$ first converges to the PRMs $\{m_j\}_{j=6}^{9}$ with goal states $\{G_n\}_{n=6}^{9}$ and then converges to the PXM $m_{10}$ with goal state G (which is also a PRM, see Fig. 4 and App. D.1 for more details on these models). After planning was performed with each of these models, we evaluated the resulting output policies in the environment. Results are shown in Fig. 6a. We can indeed see that decision-time planning performs worse when $m$ converges to a PRM, which illustrates Theoretical Result 1. To see if similar results would hold with approximators that have good generalization capabilities, we also performed the same experiment with state aggregation used in the value estimator representation. Results in Fig. 6b show that a similar trend holds in this case as well, validating Hypothesis 1.

**Common Scenario.** Theoretical Result 2 states that, when tabular value estimators are used in the simplest instantiations decision-time planning is guaranteed to perform on par or worse than background planning if the model of background planning converges to a PXM. For empirical illustration, we initialized the tabular models of both planning styles as randomized models and let them be updated through interaction to become PXMs. After every episode, we evaluated the resulting output policies in the environment. Results in Fig. 6c show that, as expected, decision-time planning performs worse after the model of background planning converges to a PXM, which illustrates Theoretical Result 2. Again, to see if similar results would hold with approximators that have good generalization capabilities, we also performed experiments with state aggregation used in the value estimator representation. Results in Fig. 6d show that a similar trend holds in this case as well, validating Hypothesis 2.

### 6.1.2 Transfer Learning Experiments

**Adaptation Scenario.** Theoretical Result 3 states that the theoretical results of the planning & learning setting would hold directly in the considered adaptation setting. For illustration, we performed an experiment

that is similar to the one in the planning & learning setting, in which we initialized the tabular models of both planning styles as randomized models and let them be updated to become PXMs. However, differently, after 25 episodes, we now added a subsequent test task to the training task in which the agent spawns in state S and has to reach the goal state $G_1$ (see App. D.1 for the details). In Fig. 6e, we can see that, similar to the planning & learning setting, before the task changes, decision-time planning performs worse after the model of background planning converges to a PXM, and the same happens after the task changes, illustrating Theoretical Result 3. Results in Fig. 6f show that a similar trend also holds when state aggregation is used in the value estimator representation, validating Hypothesis 3.

## 6.2 Experiments with Modern Instantiations

We now perform experiments with the modern instantiations of decision-time and background planning to empirically illustrate and validate our theoretical results and hypotheses in Sec. 5.2. For the experiments with the Simple Gridworld environment, we consider the same scenario in Sec. 6.1, and for the experiments with the MiniGrid, Atari and Procgen environments, we consider the scenario in which both the value estimators and models are represented with neural networks. More on the implementation details of these instantiations can be found in App. D.2.

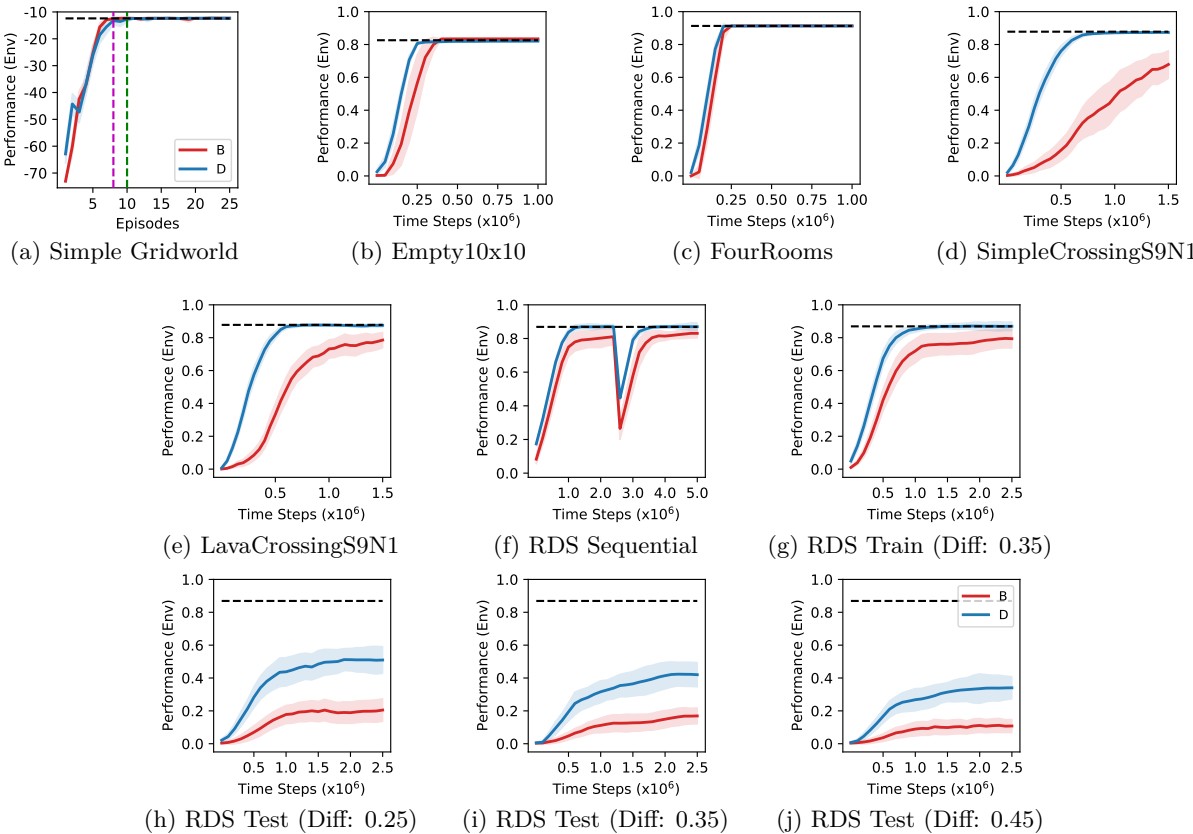

Figure 7: The performance of the modern instantiations of decision-time (D) and background (B) planning in the (a-e) planning & learning and (f-j) transfer learning settings with (a) tabular and (b-j) neural network value estimator representations. The black dashed lines indicate the performance of the optimal policy in the corresponding environment. The green and magenta dashed line in (a) inidicates the point after which decision-time and background planning's models become and remain as PXMs, respectively. Shaded regions are one standard error over (a) 250 and (b-j) 100 runs.

### 6.2.1 Planning & Learning Experiments

**Simplified Scenario.** Theoretical Result 4 states that when tabular value estimators are used in the modern instantiations, decision-time planning will now catch up with the performance background planning if both their models converge to PXMs. To illustrate, we implemented the tabular versions of the decision-time and background planning algorithms in Zhao et al. (2021) (see Alg. 5 & 6) and compared them on the Simple Gridworld environment. Results are shown in Fig. 7a. As expected,decision-time planning indeed catches up with the performance background planning when their models converge to PXMs, illustrating Theoretical Result 4.

**Original Scenario.** We then argued that the use of "hallucinated observations" in the updates of the value estimator can prevent background planning from reaching optimal or good performance. In Hypothesis 4, we hypothesized that when neural networks are used in the representation of the models we would expect deviations from the simple scenario and decision-time planning would perform better than background planning. To validate this hypotheses, we compared the decision-time and background planning algorithms in Zhao et al. (2021) (see Alg. 7 & 8) on four MiniGrid environments: Empty 10x10, FourRooms, SimpleCrossingS9N1 and LavaCrossingS9N1. The results, in Fig. 7b-7e, show that while background planning performs optimally and similar to decision-time planning in easy-to-model environments as Empty 10x10 and FourRooms, it performs suboptimally and worse than decision-time planning in hard-to-model ones as SimpleCrossingS9N1 and LavaCrossingS9N1 in which "hallucinated observations" are more of an issue.

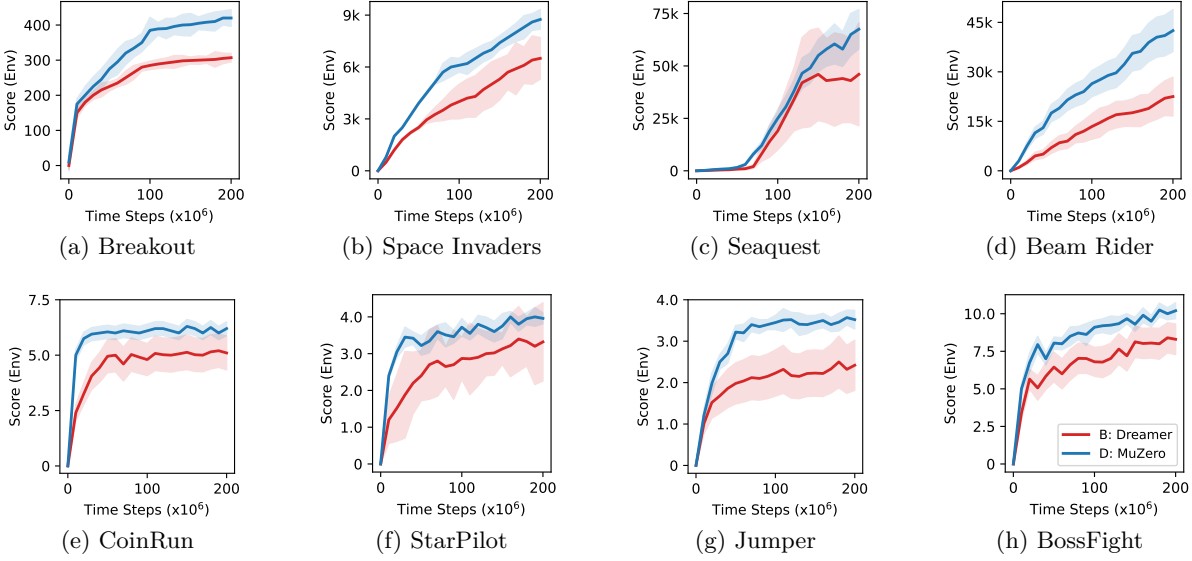

Figure 8: The performance of MuZero and DreamerV3 in the (a-d) planning & learning and (e-h) transfer learning settings. The plots are obtained by periodically evaluating the two algorithms throughout the training process. In the Procgen environments, evaluation is done on the test tasks. Shaded regions are one standard error over 5 runs.

To further test the validity of Hypothesis 4 with state-of-the-art algorithms and more complex domains, we compared MuZero (Schrittwieser et al., 2020) and DreamerV3 (Hafner et al., 2023) on four commonly-used Atari environments: Breakout, Space Invaders, Seaquest and Beam Rider. Results in Fig. 8a-8d display a similar performance trend to the results we obtained with the algorithms in Zhao et al. (2021) and MiniGrid envriondments, further validating Hypothesis 4.

### 6.2.2 Transfer Learning Experiments

In Sec. 5.2.2, we first hypothesized in Hypothesis 5 that in both the adaptation and zero-shot scenarios, we would expect decision-time planning would perform better than background planning on the training

tasks. Then, we hypothesized in Hypothesis 6 that in the adaptation scenario, we would expect background planning to suffer more in adaptation process and perform worse than decision-time planning on the test tasks. Finally, we hypothesized in Hypothesis 7 that, under certain conditions, we would expect decision-time planning to improve upon its existing policy and perform better than background planning on the test tasks.

**Adaptation Scenario.** In order to test the validity of Hypothesis 5 and 6, we compared the decision-time and background planning algorithms in Zhao et al. (2021) (see Alg. 7 & 8) on a sequential version of the RandDistShift (RDS, Zhao et al., 2021) environment. In this environment, the agent is first trained on training tasks with difficulty 0.35 (see Fig. 5e) and then it is left for adaptation to the test tasks with difficulty 0.35 (see Fig. 5g). Results in Fig. 7f show that (i) similar to the original scenario in the planning & learning setting, background planning performs suboptimally and worse than decision-time planning on the training tasks, and (ii) it indeed suffers more in the adaptation process and performs worse than decision-time planning on the test tasks, validating Hypothesis 5 and 6.

**Zero-shot Scenario.** In order to test the validity of Hypothesis 5 and 7, we again compared the decision-time and background planning algorithms in Zhao et al. (2021) (see Alg. 7 & 8) on the original RDS environment Zhao et al. (2021). In this environment, the agent is trained on training tasks with difficulty 0.35 (see Fig. 5e) and during the training process it is periodically evaluated on the test tasks with difficulties varying from 0.25 to 0.45 (see Fig. 5f-5h). Results are shown in Fig. 7g-7j. As can be seen in Fig. 7g, background planning again performs suboptimally and worse than decision-time planning on the training tasks, validating Hypothesis 5. We can also see in Fig. 7h-7j that decision-time planning indeed achieves significantly better zero-shot performance than background planning across all test tasks with varying difficulties, validating Hypothesis 7.

To test the validity of Hypothesis 7 with state-of-the-art algorithms and more complex domains, we compared MuZero and DreamerV3 on four commonly-used Procgen environments (hard difficulty, 500 train levels): CoinRun, StarPilot, Jumper, BossFight. Results in Fig. 8e-8h display a similar performance trend to the results we obtained with the algorithms in Zhao et al. (2021) and the RDS environment, validating Hypothesis 7.

## 7 Related Work

The abstract view of the two planning styles that we provide in this study is mostly related to the recent monograph of Bertsekas (2021) in which the recent successes of AlphaZero (Silver et al., 2018), a decision-time planning algorithm, are viewed through the lens of dynamic programming. However, we take a broader perspective and provide a unified view that encompasses both decision-time and background planning algorithms. Also, instead of assuming the availability of an exact model, we consider scenarios in which a model has to be learned by pure interaction with the environment. Another closely related study is the study of Hamrick et al. (2021) which informally relates MuZero (Schrittwieser et al., 2020), another decision-time planning algorithm, to various other decision-time and background planning algorithms in the literature. Our study can be viewed as a study that formalizes the relation between the two planning styles.

On the performance comparison side, there have also been benchmarking studies that empirically compare the performances of various decision-time and background planning algorithms on continuous control domains in the planning & learning setting (Wang et al., 2019), and on MiniGrid environments in specific transfer learning settings (Zhao et al., 2021). However, none of these studies provide a general understanding of when will one planning style perform better than the other. Also, rather than comparing the algorithms using the expected discounted return, these studies perform the comparison using the expected undiscounted return, and thus might be misleading in understanding the degree of optimality of the generated output policies.

Finally, our work also has connections to the studies of Jiang et al. (2015) and Arumugam et al. (2018) which provide upper bounds for the performance difference between policies generated as a result of planning with an exact and an estimated model. However, rather than providing upper bounds, in this study, we are interested in understanding which classes of models will allow for one planning style to perform better than the other. Lastly, another related line of research is the recent studies of Grimm et al. (2020; 2021) which classify models according to how relevant they are for value-based planning. Although, we share the same

Table 1: Summary of how the simplest and modern instantiations two planning styles would compare against each other across different settings. Read from left to right in a top-down fashion.

| Setting | Simplest Instantiations | Modern Instantiations |
|---------|------------------------|----------------------|
| Planning & Learning | **1) Fairest Scenario**

Background planning performs better when their model converges to a PRM (or a PXM)

**Reason:** While decision-time planning corresponds to performing one-step PI, background planning corresponds to performing full PI

**2) Common Scenario**

Background planning performs better when its model converges to a PXM

**Reason:** Due to the same reason as in the fairest scenario | **1) Simplified Scenario**

The performance gap between the two planning styles that exists in the simplest instantiations reduces and it gradually closes after the models of both planning styles converge to PXMs

**Reason:** Decision-time planning now corresponds to performing more than one-step PI and it makes use of an improving base policy

**2) Original Scenario**

Decision-time planning performs better even if the models of both planning styles converge to PXMs

**Reason:** Background planning suffers from updating its value estimator with "hallucinated observations", which can prevent it from reaching optimal or good performance |
| Transfer Learning | **1) Adaptation Scenario**

Similar to the common scenario of the planning & learning setting, background planning performs better when its model converges to a PXM

**Reason:** Due to the same reason in the planning & learning setting | **1) Adaptation Scenario**

Decision-time planning performs better even if the models of both planning styles converge to PXMs, both before and after the switch of tasks

**Reason:** Before the switch of tasks, it is due to the same reason in the original scenario of the planning & learning setting

After the switch of tasks, in addition to the reason in the planning & learning setting, it is also due to the "halluicanted" experience of background planning that resembles the training tasks, which would lead to harmful updates to its value estimator

**2) Zero-shot Scenario**

Decision-time planning performs better

**Reason:** Decision-time planning would be able to improve upon its existing policy by performing online planning |

overall idea that models should only be judged for how useful they are in the planning process, our work differs in that we classify models according to how useful they are in comparing the two planning styles.

# 8 Conclusion and Discussion

To summarize, we performed a unified analysis of decision-time and background planning and attempted to answer the following question:

> *Using the discounted return as the performance measure, under what conditions and in which settings will one planning style perform better than the other?*

In our analysis, we have tried to be as independent as possible from the specific algorithms within each planning style and tried to focus on the general working principles of them. Overall, our findings, summarized in Table 1, suggest that even though the simplest instantiations of decision-time planning do not perform as well as the simplest instantiations of background planning, the modern instantiations of it can perform

on par or better than their background planning counterparts in both the planning & learning and transfer learning settings.

We note that the main purpose of this study was to contribute towards the goal of providing a *general understanding* of under what conditions and in which settings will one planning style perform better than the other through studying the generic algorithms in their corresponding classes, and *not* to provide a benchmark that compares state-of-the-art model-based RL algorithms across various settings and domains. We also note that even though providing practical insights is not the main goal of this study at the moment, we believe that our study can guide the community in improving background planning in potentially interesting ways. For example, a possible improvement to modern background planning algorithms could be to add a meta-level algorithm that controls the usage of simulated data throughout the training process. Finally, note that we were only interested in comparing the two planning styles in terms of the expected discounted return of their output policies. Though not the main focus of this study, other possible interesting comparison directions include comparing the two planning styles in terms of their sample efficiency and real-time performance.

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

## A    Discussion on the Combined View of the Parametric and Non-Parametric Models

In order to be able to view the decision-time and background planning algorithms that perform planning with both a parametric (usually a neural network) and non-parametric (usually a replay buffer) model through our proposed unified framework, we view the two separate models of these algorithms as a single combined model and refer to it as simply a model in the main part of the paper. This becomes obvious for background planning algorithms if one notes that they perform planning with a batch of data that is jointly generated by both a parametric and non-parametric model (see e.g., line 20 in Alg. 8 in which $\phi_\theta$ and $Q_\eta$ are updated with a batch of data that is jointly generated by both $m_{b\omega}$ and $\mathcal{D}$), which can be thought of performing planning with a batch of data that is generated by a single combined model. It also becomes obvious for decision-time planning algorithms if one notes that they perform planning by first performing search with a parametric model, and then by bootstrapping on the value estimates of a continually improving policy that is obtained by planning with a non-parametric model (see e.g., line 13 in Alg. 7 in which action selection is done with both $m_{d\omega}$ and $Q_\eta$ (which is obtained by planning with $\mathcal{D}$)), which can be thought of performing planning with a single combined model that is obtained by concatenating the parametric and non-parametric models.

## B    Discussion on the Choice of the Simplest Instantiations of the Two Planning Styles

As indicated in the main paper, for decision-time planning we study the online Monte-Carlo planning (OMCP) algorithm of Tesauro & Galperin (1996), and for background planning we study the Dyna-Q algorithm of Sutton (1990; 1991). We choose these algorithms as they are the simplest instantiations in their corresponding classes and they are easy to analyze. In this study, as we are interested in scenarios where the model has to be learned from pure interaction, we consider a version of the OCMP algorithm in which the model is learned from experience (see Alg. 1 for the pseudocode). Note that this is the only difference compared to the original version of the OMCP algorithm proposed in Tesauro & Galperin (1996). And, in order to make a fair comparison with this version of the OMCP algorithm, we consider a simplified version of the Dyna-Q algorithm (see Alg. 3 & 4 for the pseudocodes of the original and simplified versions, respectively). Compared to the original version of Dyna-Q, in this version, there are several minor differences:

- While planning, the agent can now sample states and actions that it has not observed or taken before. Note that this is also the case for the OMCP algorithm considered in this study.

- Now, instead of using samples from both the environment and model, the agent updates its value estimator with samples only from the model. Note that the OMCP algorithm also makes use of only the model while performing planning.

---

**Algorithm 1** Tabular Online Monte-Carlo Planning with an adaptable model

1: Initialize $\pi^i \in \Pi$ as a random policy
2: Initialize $m_d(s,a) \; \forall s \in \mathcal{S} \; \& \; \forall a \in \mathcal{A}$
3: $n_r \leftarrow$ number of episodes to perform rollouts
4: **while** $m_d$ has not converged **do**
5:     $S \leftarrow$ reset environment
6:     **while** not done **do**
7:         $A \leftarrow \epsilon\text{-greedy}(\text{MC\_rollout}(S, m_d, n_r, \pi^i))$
8:         $R, S', \text{done} \leftarrow \text{environment}(A)$
9:         Update $m_d(S, A)$ with $R, S', \text{done}$
10:         $S \leftarrow S'$
11:     **end while**
12: **end while**
13: **Return** $m_d(s,a)$

---

**Algorithm 2** Tabular Exhaustive Search (Campbell et al., 2002) with an adaptable model

1: Initialize $m_d(s,a) \; \forall s \in \mathcal{S} \; \& \; \forall a \in \mathcal{A}$
2: $h \leftarrow$ search heuristic
3: **while** $m_d$ has not converged **do**
4:     $S \leftarrow$ reset environment
5:     **while** not done **do**
6:         $A \leftarrow \epsilon\text{-greedy}(\text{exhaustive\_search}(S, m_d, h))$
7:         $R, S', \text{done} \leftarrow \text{environment}(A)$
8:         Update $m_d(S, A)$ with $R, S', \text{done}$
9:         $S \leftarrow S'$
10:     **end while**
11: **end while**
12: **Return** $m_d(s,a)$
13:

---

**Algorithm 3** Tabular Dyna-Q

1: Initialize $Q(s,a)$ $\forall s \in \mathcal{S}$ & $\forall a \in \mathcal{A}$
2: Initialize $m_b(s,a)$ $\forall s \in \mathcal{S}$ & $\forall a \in \mathcal{A}$
3: $\mathcal{SA}_{\text{prev}} \leftarrow \{\}$
4: $n_p \leftarrow$ number of time steps to perform planning
5: **while** $Q$ and $m_b$ has not converged **do**
6:     $S \leftarrow$ reset environment
7:     **while** not done **do**
8:         $A \leftarrow \epsilon$-greedy$(Q(S,\cdot))$
9:         $R, S', \text{done} \leftarrow$ environment$(A)$
10:        $\mathcal{SA}_{\text{prev}} \leftarrow \mathcal{SA}_{\text{prev}} + \{(S,A)\}$
11:        Update $Q(S,A)$ with $R, S', \text{done}$
12:        Update $m_b(S,A)$ with $R, S', \text{done}$
13:        $i \leftarrow 0$
14:        **while** $i < n_p$ **do**
15:           $S_{m_b}, A_{m_b} \leftarrow$ sample from $\mathcal{SA}_{\text{prev}}$
16:           $R_{m_b}, S'_{m_b}, \text{done}_{m_b} \leftarrow m_b(S_{m_b}, A_{m_b})$
17:           Update $Q(S_{m_b}, A_{m_b})$ with $R_{m_b}, S'_{m_b}, \text{done}_{m_b}$
18:           $i \leftarrow i + 1$
19:        **end while**
20:        $S \leftarrow S'$
21:     **end while**
22: **end while**
23: **Return** $Q(s,a)$

**Algorithm 4** Tabular Dyna-Q of interest

1: Initialize $Q(s,a)$ $\forall s \in \mathcal{S}$ & $\forall a \in \mathcal{A}$
2: Initialize $m_b(s,a)$ $\forall s \in \mathcal{S}$ & $\forall a \in \mathcal{A}$
3: $n_p \leftarrow$ number of time steps to perform planning
4: **while** $Q$ and $m_b$ has not converged **do**
5:     $S \leftarrow$ reset environment
6:     **while** not done **do**
7:         $A \leftarrow \epsilon$-greedy$(Q(S,\cdot))$
8:         $R, S', \text{done} \leftarrow$ environment$(A)$
9:         Update $m_b(S,A)$ with $R, S', \text{done}$
10:        $i \leftarrow 0$
11:        **while** $i < n_p$ **do**
12:           $S_{m_b}, A_{m_b} \leftarrow$ sample from $\mathcal{S} \times \mathcal{A}$
13:           $R_{m_b}, S'_{m_b}, \text{done}_{m_b} \leftarrow m_b(S_{m_b}, A_{m_b})$
14:           Update $Q(S_{m_b}, A_{m_b})$ with $R_{m_b}, S'_{m_b}, \text{done}_{m_b}$
15:           $i \leftarrow i + 1$
16:        **end while**
17:        $S \leftarrow S'$
18:     **end while**
19: **end while**
20: **Return** $Q(s,a)$
21:
22:
23:

## C   Proofs

**Proposition 1.** *Let $m \in \mathcal{M}$ be a PRM of $m^*$ with respect to $\Pi = \{\pi_m^r, \pi_m^{ce}\} \subseteq \mathbb{\Pi}$ and $J$. Then, $J_{m^*}^{\pi_m^{ce}} \geq J_{m^*}^{\pi_m^r}$.*

*Proof.* This result directly follows from Defn. 2 & 5. Recall that, according to Defn. 5, given a $\pi^b \in \mathbb{\Pi}$, $\pi_m^r$ and $\pi_m^{ce}$ are the policies that are obtained after performing one-step PI and full PI in model $m$, respectively. Thus, we have $J_m^{\pi_m^r} \leq J_m^{\pi_m^{ce}}$ (Bertsekas & Tsitsiklis, 1996), which, by Defn. 2, implies $J_{m^*}^{\pi_m^{ce}} \geq J_{m^*}^{\pi_m^r}$.    □

**Proposition 2.** *Let $m_d \in \mathcal{M}$ be any model of $m^*$ with respect to $\Pi_d = \{\pi_{m_d}^r, \pi_{m_d}^{ce}\} \subseteq \mathbb{\Pi}$ and $J$, and let $m_b \in \mathcal{M}$ be a PXM of $m^*$ with respect to $\Pi_b = \{\pi_{m_b}^r, \pi_{m_b}^{ce}\} \subseteq \mathbb{\Pi}$ and $J$. Then, $J_{m^*}^{\pi_{m_b}^{ce}} \geq J_{m^*}^{\pi_{m_d}^r}$.*

*Proof.* This result directly follows from Defn. 4 & 5. Recall that, according to Defn. 5, given a $\pi^b \in \mathbb{\Pi}$, $\pi_{m_b}^{ce}$ is the policy that is obtained after performing full PI in model $m_b$. Thus, $\pi_{m_b}^{ce}$ is one of the optimal policies of model $m_b$ (Bertsekas & Tsitsiklis, 1996), which, by Defn. 4, implies $J_{m^*}^{\pi_{m_b}^{ce}} = \max_{\pi \in \Pi} J_{m^*}^{\pi}$ and thus $J_{m^*}^{\pi_{m_b}^{ce}} \geq J_{m^*}^{\pi} \forall \pi \in \mathbb{\Pi}$. This in turn implies $J_{m^*}^{\pi_{m_b}^{ce}} \geq J_{m^*}^{\pi_{m_d}^r}$.    □

## D   Experimental Details

In this section, we provide the details of the experiments that are performed in Sec. 6. This also includes the implementation details of the simplest and modern instantiations of the two planning styles that are considered in this study. In all of the experiments on the Simple Gridworld environment we have calculated the performance with a discount factor of 0.9, and in all of the experiments with the MiniGrid environments we have calculated it with a discount factor of 0.99.

### D.1   Details of the Simplest Instantiation Experiments

### D.1.1   Environments & Models

All of the experiments in Sec. 6.1 are performed on the Simple Gridworld environment. Here, as explained in Sec. 4, the agent spawns in state S and has to navigate to the goal state depicted by G. At each time step, the agent receives an $(x, y)$ pair indicating its position, and based on this, selects an action that moves it

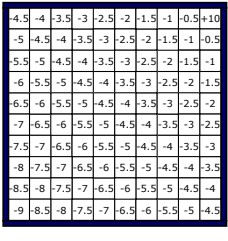
(a) Simple Gridworld

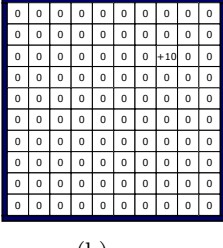
(b) $m_8$

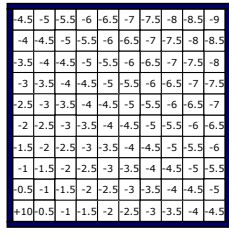
(c) Subsequent Test Task


(d) State Aggregation

Figure D.1: (a, b, c) Reward functions of (a) the Simple Gridworld environment, (b) the $m_8$ model and (c) the subsequent test task in the adaptation scenario. (d) The form of state aggregation used in this study, in which four neighboring cells are grouped into a single cell.

to one of the four neighboring cells with a slip probability of 0.05. The agent receives a negative reward that is linearly proportional to its distance from G and a reward of +10 if it reaches G (see Fig. D.1a). The agent-environment interaction lasts for a maximum of 100 time steps and after this the episode terminates with a reward of 0 if the agent was not able to reach the goal state G.

Further details of the environments and models that are used in the planning & learning and transfer learning experiments are as follows:

- **Planning & Learning Setting**
  - **Fairest Scenario**
    For the experiments in the fairest scenario, $m$ was trained to sequentially converge to a set of PRMs in which the agent receives a reward of +10 if it reaches the goal state and a reward of 0 elsewhere. For example, see the reward function of model $m_8$ in Fig. D.1b. Note that these models have the same transition distribution and initial state distribution with the Simple Gridworld environment.
  - **Common Scenario**
    For the experiments in the common scenario, we have assumed that the agent already has access to the transition distribution and initial state distribution of the environment, and only has to learn the reward function.

- **Transfer Learning Setting**
  - **Adaptation Scenario**
    Finally, for the experiments in the adaptation scenario, we considered a subsequent test task with a reward function as in Fig. D.1c, which is a transposed version of the training task's reward function (Fig. D.1a). Note again that we have assumed that the agent already has access to the transition distribution and initial state distribution of the environment, and only has to learn the reward function.

### D.1.2 Implementation Details of the Simplest Instantiations

For our simplest instantiation experiments, we considered the versions of the OMCP (Tesauro & Galperin, 1996) and the Dyna-Q (Sutton, 1990; 1991) algorithms described in Sec. B. The pseudocodes of these algorithms are presented in Alg. 1 & 4, respectively, and the details of them are provided in Table D.1 & D.2, respectively. For our function approximation experiments, we have used a state aggregator of the form in Fig. D.1d.

Table D.1: Details and hyperparameters of Alg. 1.

| | |
|---|---|
| $\pi^i$ | deterministic random policy |
| $m_d$ | tabular model |
| $n_r$ | 50 |
| $\epsilon$ | linearly decays from 1.0 to 0.0 over 20 episodes |

Table D.2: Details and hyperparameters of Alg. 4.

| | |
|---|---|
| $Q$ | tabular value function (initialized as zero everywhere) |
| $m_b$ | tabular model |
| $n_p$ | 100 |
| $\epsilon$ | linearly decays from 1.0 to 0.0 over 20 episodes |

### D.2 Details of the Modern Instantiation Experiments

### D.2.1 Environments & Models

In Sec. 6.2, we performed experiments on four different domains. The details of these environments and their corresponding models across the different settings are as follows:

- **Planning & Learning Setting**

    - **Simplest Scenario**
        * **Simple Gridworld Environment**
        We refer the reader to Sec. D.1 for the details of the Simple Gridworld environment as we have used the same environment in the modern instantiation experiments as well. To learn about the models of both planning styles, we also refer the reader to Sec. D.1 as have used the same models in the modern instantiation experiments as well.

    - **Original Scenario**
        * **MiniGrid Environments**
        We performed experiments on the Empty 10x10, FourRooms, SimpleCrossingS9N1 and LavaCrossingS9N1 environments (see Fig. 5a-5d). While the last two of these environments already pre-exist in MiniGrid, the first two of them are manually built environments. Specifically, (i) the Empty 10x10 environment is obtained by expanding the Empty 8x8 environment to a size of 10x10 and (ii) the FourRooms environment is obtained by contracting the 16x16 FourRooms environment to a size of 10x10. More on the details of these environments can be found in Chevalier-Boisvert et al. (2018).

        * **Atari Environments**
        We performed experiments on the Breakout, Space Invaders, Seaquest and Beam Rider environments. More on the details of these environments can be found in Bellemare et al. (2013).

- **Transfer Learning Setting**

    - **Adaptation Scenario**
        * **MiniGrid Environments**
        We performed experiments on the sequential version of the RandDistShift (RDS) environment that was introduced in Zhao et al. (2021) (referred to as RDS Sequential). In the main article, we have already provided the necessary details of this environment. Additionally, we note that in our adaptation experiments, we reinitialized the replay buffers of both planning styles after the tasks switch from the training tasks to the test tasks.

    - **Zero-shot Scenario**
        * **MiniGrid Environments**
        We performed experiments on the regular version of the RDS environment(Zhao et al., 2021) (see Fig. 5e-5h). In the main article, we have already provided the necessary details of this environment. Readers who are interested in learning more about the details can refer to the study of Zhao et al. (2021).

        * **Procgen Environments**
        We performed experiments on the CoinRun, StarPilot, Jumper and BossFight environments. For evaluation, we have used the standard protocol of training on 500 training tasks and testing on an infinite number of procedurally generated test tasks (hard difficulty, 500 train levels). More on the details of these environments can be found in Cobbe et al. (2020).

Finally, note that, as opposed to our Simple Gridworld experiments, in our experiments with the MiniGrid, Atari and Procgen environments, we did not enforce any kind of structure on the models of the agent and just initialized them randomly.

### D.2.2 Implementation Details of the Modern Instantiations

For our modern instantiation experiments, we first performed experiments with the tabular versions of the decision-time and background planning algorithms in Zhao et al. (2021), whose pseudocodes are presented in Alg. 5 & 6, respectively. The details of these algorithms are provided in Table D.3 & D.4, respectively.

Table D.3: Details and hyperparameters of Alg. 5.

| $Q$ | tabular value function (initialized as zero everywhere) |
|---|---|
| $m_d$ | tabular model |
| $n_s$ | $|\mathcal{A}|$ |
| $h$ | breadth-first search |
| $\epsilon$ | linearly decays from 1.0 to 0.0 over 20 episodes |

Table D.4: Details and hyperparameters of Alg. 6.

| $Q$ | tabular value function (initialized as zero everywhere) |
|---|---|
| $m_b$ | tabular parametric model |
| $n_p$ | 50 |
| $\epsilon$ | linearly decays from 1.0 to 0.0 over 20 episodes |

---

**Algorithm 5** The tabular version of the Decision-Time Planning algorithm in Zhao et al. (2021)

1: Initialize $Q(s, a)$ $\forall s \in \mathcal{S}$ & $\forall a \in \mathcal{A}$
2: Initialize $m_d(s, a)$ $\forall s \in \mathcal{S}$ & $\forall a \in \mathcal{A}$
3: Initialize the replay buffer $\mathcal{D} \leftarrow \{\}$
4: $n_s \leftarrow$ number of time steps to perform search
5: $h \leftarrow$ search heuristic
6: **while** $m_d$ and $\mathcal{D}$ has not converged **do**
7:      $S \leftarrow$ reset environment
8:      **while** not done **do**
9:          $A \leftarrow \epsilon$-greedy(search_with_bootstrap$(S, m_d, Q, n_s, h)$)
10:          $R, S', \text{done} \leftarrow$ environment$(A)$
11:          $\mathcal{D} \leftarrow \mathcal{D} + \{(S, A, R, S', \text{done})\}$
12:          $S_{\mathcal{D}}, A_{\mathcal{D}}, R_{\mathcal{D}}, S'_{\mathcal{D}}, \text{done}_{\mathcal{D}} \leftarrow$ sample from $\mathcal{D}$
13:          Update $Q$ & $m_d$ with $S_{\mathcal{D}}, A_{\mathcal{D}}, R_{\mathcal{D}}, S'_{\mathcal{D}}, \text{done}_{\mathcal{D}}$
14:          $S \leftarrow S'$
15:      **end while**
16: **end while**
17: **Return** $Q$ & $m_d(s, a)$
18:
19:
20:
21:
22:
23:
24:

---

**Algorithm 6** The tabular version of the Background Planning algorithm in Zhao et al. (2021)

1: Initialize $Q(s, a)$ $\forall s \in \mathcal{S}$ & $\forall a \in \mathcal{A}$
2: Initialize $m_b(s, a)$ $\forall s \in \mathcal{S}$ & $\forall a \in \mathcal{A}$
3: Initialize the replay buffer $\mathcal{D} \leftarrow \{\}$
4: $n_p \leftarrow$ number of time steps to perform planning
5: **while** $Q$, $m_b$ and $\mathcal{D}$ has not converged **do**
6:      $S \leftarrow$ reset environment
7:      **while** not done **do**
8:          $A \leftarrow \epsilon$-greedy$(Q(S, \cdot))$
9:          $R, S', \text{done} \leftarrow$ environment$(A)$
10:          Update $m_b(S, A)$ with $R$, $S'$, done
11:          $\mathcal{D} \leftarrow \mathcal{D} + \{(S, A, R, S', \text{done})\}$
12:          $i \leftarrow 0$
13:          **while** $i < n_p$ **do**
14:              $S_{m_b}, A_{m_b} \leftarrow$ sample from $\mathcal{S} \times \mathcal{A}$
15:              $R_{m_b}, S'_{m_b}, \text{done}_{m_b} \leftarrow m_b(S_{m_b}, A_{m_b})$
16:              Update $Q(S_{m_b}, A_{m_b})$ with $R_{m_b}, S'_{m_b}, \text{done}_{m_b}$
17:              $S_{\mathcal{D}}, A_{\mathcal{D}}, R_{\mathcal{D}}, S'_{\mathcal{D}}, \text{done}_{\mathcal{D}} \leftarrow$ sample from $\mathcal{D}$
18:              Update $Q(S_{\mathcal{D}}, A_{\mathcal{D}})$ with $R_{\mathcal{D}}$, $S'_{\mathcal{D}}$, $\text{done}_{\mathcal{D}}$
19:              $i \leftarrow i + 1$
20:          **end while**
21:          $S \leftarrow S'$
22:      **end while**
23: **end while**
24: **Return** $Q(s, a)$

Then, we performed experiments with the decision-time and background planning algorithms in Zhao et al. (2021). More specifically, for decision-time planning we study the "UP" algorithm, and for background planning we study the "Dyna" algorithm in Zhao et al. (2021).[13] The pseudocodes of these algorithms are presented in Alg. 7 & 8, respectively, and the details of them are provided in Table D.5 & D.6, respectively. Note that we have kept the details and the hyperparamters the same as Zhao et al. (2021). For more information on the details such as the neural network architectures, replay buffer sizes, learning rates, exact details of the tree search …, we refer the reader to the publicly available code[14] and the supplementary material of Zhao et al. (2021).

---

[13] Note that these two algorithms do not employ the "bottleneck mechanism" introduced in Zhao et al. (2021).

[14] See `https://github.com/mila-iqia/Conscious-Planning` for the publicly available code.

---

**Algorithm 7** The Decision-Time Planning algorithm in Zhao et al. (2021)

---

1: Initialize the parameters $\theta$, $\eta$ & $\omega$ of $\phi_\theta : \mathcal{O}_E \to \mathcal{S}_A$, $Q_\eta : \mathcal{S}_A \times \mathcal{A}_E \to \mathbb{R}$ & $m_{d\omega} = (p_\omega, r_\omega, d_\omega)$
2: Initialize the replay buffer $\mathcal{D} \leftarrow \{\}$
3: $N_{ple} \leftarrow$ number of episodes to perform planning and learning
4: $N_{rbt} \leftarrow$ number of samples that the replay buffer must hold to perform planning and learning
5: $n_s \leftarrow$ number of time steps to perform search
6: $n_{bs} \leftarrow$ number of samples to sample from $\mathcal{D}$
7: $h \leftarrow$ search heuristic
8: $S \leftarrow$ replay buffer sampling strategy
9: $i \leftarrow 0$
10: **while** $i < N_{ple}$ **do**
11:     $O \leftarrow$ reset environment
12:     **while** not done **do**
13:         $A \leftarrow \epsilon$-greedy(tree_search_with_bootstrapping($\phi_\theta(O), m_{d\omega}, Q_\eta, n_s, h$))
14:         $R, O', \text{done} \leftarrow \text{environment}(A)$
15:         $\mathcal{D} \leftarrow \mathcal{D} + \{(O, A, R, O', \text{done})\}$
16:         **if** $|\mathcal{D}| \geq N_{rbt}$ **then**
17:             $\mathcal{B} \leftarrow \text{sample\_batch}(\mathcal{D}, n_{bs}, S)$
18:             Update $\phi_\theta$, $Q_\eta$ & $m_{d\omega}$ with $\mathcal{B}$
19:         **end if**
20:         $O \leftarrow O'$
21:     **end while**
22:     $i \leftarrow i + 1$
23: **end while**
24: **Return** $\phi_\theta$, $Q_\eta$ & $m_{d\omega}$

---

**Algorithm 8** The Background Planning algorithm in Zhao et al. (2021)

---

1: Initialize the parameters $\theta$, $\eta$ & $\omega$ of $\phi_\theta : \mathcal{O}_E \to \mathcal{S}_A$, $Q_\eta : \mathcal{S}_A \times \mathcal{A}_E \to \mathbb{R}$ & $m_{b\omega} = (p_\omega, r_\omega, d_\omega)$
2: Initialize the replay buffer $\mathcal{D} \leftarrow \{\}$ and the imagined replay buffer $\mathcal{D}_i \leftarrow \{\}$
3: $N_{ple} \leftarrow$ number of episodes to perform planning and learning
4: $N_{rbt} \leftarrow$ number of samples that the replay buffer must hold to perform planning and learning
5: $n_{ibs} \leftarrow$ number of samples to sample from $\mathcal{D}_i$
6: $n_{bs} \leftarrow$ number of samples to sample from $\mathcal{D}$
7: $S \leftarrow$ replay buffer sampling strategy
8: $i \leftarrow 0$
9: **while** $i < N_{ple}$ **do**
10:     $O \leftarrow$ reset environment
11:     **while** not done **do**
12:         $A \leftarrow \epsilon$-greedy($Q_\eta(\phi_\theta(O), \cdot)$)
13:         $R, O', \text{done} \leftarrow \text{environment}(A)$
14:         $\mathcal{D} \leftarrow \mathcal{D} + \{(O, A, R, O', \text{done})\}$
15:         $\mathcal{D}_i \leftarrow \mathcal{D}_i + \{(\phi_\theta(O), A)\}$
16:         **if** $|\mathcal{D}| \geq N_{rbt}$ **then**
17:             $\mathcal{B}_i \leftarrow \text{sample\_batch}(\mathcal{D}_i, n_{ibs}, S)$
18:             $\mathcal{B}_i \leftarrow \mathcal{B}_i + m_{b\omega}(\mathcal{B}_i)$
19:             $\mathcal{B} \leftarrow \text{sample\_batch}(\mathcal{D}, n_{bs}, S)$
20:             Update $\phi_\theta$ & $Q_\eta$ with $\mathcal{B}_i + \mathcal{B}$
21:             Update $\phi_\theta$ & $m_{b\omega}$ with $\mathcal{B}$
22:         **end if**
23:         $O \leftarrow O'$
24:     **end while**
25:     $i \leftarrow i + 1$
26: **end while**
27: **Return** $\phi_\theta$ & $Q_\eta$

---

Table D.5: Details and hyperparameters of Alg. 7.

| | |
|---|---|
| $\phi_\theta$ | convolutional neural network |
| $Q_\eta$ | multilayer perceptron |
| $m_{d\omega}$ | multilayer perceptron |
| $N_{ple}$ | 50M |
| $N_{rbt}$ | 50k |
| $n_s$ | 5 |
| $n_{bs}$ | 128 |
| $h$ | best-first search (training), random search (evaluation) |
| $S$ | random sampling |
| $\epsilon$ | linearly decays from 1.0 to 0.0 over 1M time steps |

Table D.6: Details and hyperparameters of Alg. 8.

| | |
|---|---|
| $\phi_\theta$ | convolutional neural network |
| $Q_\eta$ | multilayer perceptron |
| $m_{b\omega}$ | multilayer perceptron |
| $N_{ple}$ | 50M |
| $N_{rbt}$ | 50k |
| $(n_{ibs}, n_{bs})$ | $(128, 128)$ |
| $S$ | random sampling |
| $\epsilon$ | linearly decays from 1.0 to 0.0 over 1M time steps |

Finally, we performed experiments with both MuZero (Schrittwieser et al., 2020) and DreamerV3 (Hafner et al., 2021), which are state-of-the-art decision-time and background planning algorithms, respectively. For MuZero we have used the open-source implementation of Niu et al. (2023) and for DreamerV3 we have used the publicly available code of Hafner et al. (2023). And, for our Atari experiments we have used the default configs and hyperparameters that were provided in Schrittwieser et al. (2020) and Hafner et al. (2023). Finally, for our Procgen experiments we have used the same configs and hyperparameters of the Atari experiments. For more information on the details, we refer the reader to studies of Schrittwieser et al. (2020) and Hafner et al. (2023).

