# OpenReview forum: "Towards an Understanding of Decision-Time vs. Background Planning in Model-Based Reinforcement Learning"
_TMLR — Rejected by TMLR_

### Review · Reviewer_JPaQ · 2023-11-06

**Summary Of Contributions:**

This paper presents a study that compares decision-time planning and background planning in model-based RL, by providing a framework based on dynamic programming and constructing multiple scenarios & setups for investigating their performances based on discounted returns. The main contribution of this paper is to design a way to compare two approaches that are often based on different implementations and subtle differences, and providing empirical analysis based on that experimental design.

**Audience:**

Yes

**Claims And Evidence:**

No

**Requested Changes:**

- Could analysis and claims based on *modern* instantiations be supported with more evidences to address my points in the Weaknesses? I think the is the most crucial for making me have a confidence in recommending the paper to be accepted.
- Could authors elaborate more on the sentence in Page 7?: `We refer to this transfer setting as the adaptation setting. In this setting, we would expect the results of the planning & learning setting to hold directly, as instead of a single one, there are now two consecutive planning & learning settings.`
- On `Thus, we hypothesize that compared to decision-time planning, it is likely for background planning to suffer more in reaching optimal (or close-to-optimal) performance in their modern instantiations` -- As the authors said, modern instantiations of background planning already contains some components to avoid the issue from using simulated data -- I'm unconvinced that removing that component for a *fair* comparison can be indeed fair.

**Strengths And Weaknesses:**

Strengths
- Authors did a really good job of explaining decision-time planning and background planning in model-based RL, and motivates well why and how comparing them can be important and interesting to the community.
- Interesting analysis based on two widely-considered setups.

Weaknesses
- Analysis using the modern instantiations is not sufficient to support the claims made in the paper. I understand the goal of this paper is not to be a benchmark paper, but the scope of the experiments in the paper is too narrow for making a concrete conclusion.
- For instance, *modern* instantiations include a lot of implementation details that have enabled them to work on more complex setups (e.g., visuomotor control, high-resolution, long-horizon video games, ..) but not considering them for a fair comparison makes the paper's analysis be underwhelming in its scope and make me question whether the conclusion is generic enough to hold in various setups. For instance, the paper could be more strengthened by reporting the strongest available performance of each approach using the implementation currently available, and also reporting the performance by ablating some components for a fair comparison. Moreover, evaluation of modern instantiations on more complex setups could be also important for making the claims made in the paper be more strongly supported.
- Writing can be improved by introducing a more structure and also providing a visualization in Section 3, as its paragraph consists of some very long sentences and is a bit difficult to parse.

---

> ### Comment · Reviewer_JPaQ · 2024-01-15
>
> Thanks for the response. This will be hopefully my final comment as it seems like almost everything is clearly delivered.
>
> > We are not quite sure by what is meant by the current results not supporting the original results as the results with MuZero and Dreamer are following the same trends with the algorithms taken from Zhao et al.
>
> - I'd rather say that these additional results are not supporting the original results because they are not controlled as the original results, hope this everything is clear on this side.
>
> > We will look into this issue and add additional experiments with PlaNet in the camera-ready version of the paper if gets accepted.
>
> - Just as a note in case this helps, I noticed [this repository](https://github.com/mazpie/mastering-urlb) contains PyTorch implementation of DreamerV2 with the MPC support and [repo2](https://github.com/nicklashansen/tdmpc) could be also useful for potential future experiments.

---

### Review · Reviewer_sGBH · 2023-11-22

**Summary Of Contributions:**

In this paper, the authors discuss two styles of planning algorithms in reinforcement learning, which are decision-time planning and background planning. Theoretical analysis and experiments on several environments are given and the benefits of the two styles are discussed accordingly.

**Audience:**

Yes

**Claims And Evidence:**

No

**Requested Changes:**

1. Discussion of relationship of two styles of planning to other existing terms in model-based reinforcement learning such as “dyna”, “world models”, “model predictive control”, “online planning”.
2. Discussion of the development related work of AlphaGo, AlphaZero, MuZero and their design choices of between decision-time planning and background planning
3. Discussion of state-of-the-art model-based planning algorithms for environments in image space such as Dreamers (v1, v2, v3). It’s worth noting that Dreamers (and most other competitors) were not using decision-time planning and it would be interesting to discuss if/how the paper could help existing algorithms like Dreamers to improve their performance.

**Strengths And Weaknesses:**

Strengths:
The paper is well written and provides an adequate related work section.

Weakness:
The paper does not have sufficient experiment sections to support the claims in this paper.
I think it will be fair to value other reviewers’ opinion on the theoretical part of the paper, since I couldn’t understand it.
My concern of the paper is based on the fact that none of the challenging benchmarks are considered. It is worth mentioning most of the current model-based planning algorithms are considering very challenging environments as hard as Minecraft or DM lab environments as studied in DreamerV3 [1], which have images as input to the network and are orders of magnitude more complex.
That being said, the paper makes very strong assumptions on the environments and the environments are too simple. This makes the conclusion of the paper less convincing.

One might even say this is detached from the reality of reinforcement learning applications.

I do understand theoretical reinforcement learning has its own value and I am not too familiar with it and I might be biased. I would like the area chair to value the reviews from other reviewers for a fairer evaluation.

[1] Hafner, Danijar, Jurgis Pasukonis, Jimmy Ba, and Timothy Lillicrap. "Mastering diverse domains through world models." arXiv preprint arXiv:2301.04104 (2023).

---

### Review · Reviewer_96Kr · 2023-12-19

**Summary Of Contributions:**

This paper has an admirable and well-aligned-with-TMLR goal of better understanding when/where/why decision-time planning performs better/worse than background planning. The paper lays out different sets of model, where each set is distinguished by an ordering of policies in the model vs the ordering of policies in the real environment. For two sets of interest, PRMs and PXMs (where policy orderings are aligned with the real env) the paper provides theoretical results showing that background planning outperforms decision-time planning. The paper investigates these findings empirically with tabular models and in simple environments where it can construct/measure these sets of interest explicitly and shows that in some settings, this finding (b > dt) holds. When extending to multistep rollout models or when utilizing simulated experience, the paper finds the ordering (b > dt) no longer holds.

**Audience:**

Yes

**Claims And Evidence:**

No

**Requested Changes:**

1. Understand, measure, and report the confounding factors that cause the conclusions to flip (b > dt, then dt > b). Is it due to compounding errors during multistep rollout?
2. Address the writing concerns listed above. Consider restructuring the document so as to avoid needing subsubsections (I don't think they helped).
3. Explicit define "modern" setting. What does this mean in the context of an "understanding" paper? Are there particular design patterns in the "modern" algorithms that are impacting results? Perhaps point 3 here is ultimately a restating of point 1.
4. Provide a lot more empirical details. State the hypotheses being tested, how the empirical design is structured, and motivate how that empirical design answers the hypothesis if it is not self-evident.

**Strengths And Weaknesses:**

## Did the paper accomplish its goal?
The paper states a very clear goal: understand under what conditions and in which settings one planning style outperforms the other. Did the paper accomplish this goal?

The theoretical results give a strong indication of several conditions where one planning style (background) clearly outperforms the other. Following this theory, however, the paper walks back the claims and makes opposing conjectures (see bottom of page 8) due to some "implementation details" of the algorithms. Through Section 4, we go on a journey from feeling like we understand the conditions where background > decision-time, to a conclusion hypothesizing that dt > b. This suggests, then, that perhaps we do not have a complete or useful understanding. Section 5, the empirical section, has a similar structure. We start from a place of complete understanding with expected results (b > dt), but conclude with flipped results (dt > b).

This contradiction is even evident in the abstract to the paper. To summarize the abstract:
1. We provide theory in the simple setting that b > dt.
2. We hypothesize that in the "modern" setting dt > b.
3. We empirically show that in the simple setting b > dt and in the modern setting sometimes dt > b.

Note that in point (2) the paper provides only hypotheses as to why dt > b; hypotheses, but not understanding. These hypotheses as to "under what conditions" and "in which settings" are not tested/validated in the empirical section. Only the dt > b hypothesis is tested.

As a whole, then, I would argue that this paper does not accomplish its stated goal of providing further understanding. It _clearly_ makes a good first step toward understanding in the simple setting, however this understanding fails to hold in the "modern" setting. It seems that the failure point occurs when the planning algorithm changes to use multistep rollouts due to compounding model error. It seems likely that a rigorous understanding is possible when this moderating variable is explicitly accounted for in the empirical study.

## Clarity
Overall, I found this paper quite difficult to read and I strongly suspect this would limit the impact of the work. I think the paper would greatly benefit from a broad-scale reorganization and better upfront symbol planning. Each of the following may individually seem pedantic or minor, but I believe this is a "death by 1000 cuts" scenario. The following are in no particular order (numbered only to make it easier to discuss if needed):

1. There is significant mental overhead for keeping track of symbols and terms in the paper. What is a $\overline{m}$ vs. a $\overline{\overline{m}}$? These two symbols do not express their intent in a self-evident way. It's clear that they are both model instances (lowercase $m$), but what do the overlines mean? Is there significance in two overlines vs. one? (these questions are rhetorical)
	1. If they were only used in the paragraph where they were defined (middle of page 7), this would be fine. But they appear again later in the paper in a different context (bottom of page 8). These could be simplified by having the symbol relate to the concept that it refers to. For instance, let's call background planning $B$ and decision-time planning $D$, then we could instead use $m_B$ and $m_D$ to refer to the same concepts. Conveniently, now we could also use $\Pi_B$ and $\Pi_D$ to refer to these sets of policies, perhaps $\pi^\text{ce}$ could instead be $\pi^B$ and $\pi^\text{r}$ could be $\pi^D$ and so on. Relatedly, what is $c$ in $m_c$?
2. I'm not sure what the >=< stacked symbol means in Def'n 1 and 2. I suspect most readers are in the same boat. Some quick footnote or comment in the text defining your use of symbols would be _very_ helpful for comprehension when using atypical symbols.
3. The text often talks about a model converging to some set (e.g. mid-page 8: "if both $\overline{m}_c$ and $\overline{\overline{m}}_c$ converge to PXMs"). However as model is defined in Sec 2 and PXM in Def'n 4, I'm not sure this rigorously makes sense. A model instance either is a PXM or isn't; it cannot converge in any meaningful sense. I suspect the paper is meaning to refer to a sequence of models generated by some learning algorithm---for which convergence of a _sequence_ makes sense---however this is not clearly stated in the paper. At the very least, $m$ needs to be subscripted by a temporal parameter (e.g. $m_t$) to imply a sequence.
4. "Convergence" is used in other contexts that don't make sense. For instance (page 6): "Models of the two planning styles converge to...". What is the relationship between _planning style_ and _model convergence_? I can consider a fixed model $m$ and discuss the difference between planning styles, which implies this relationship is not self-evident. I suspect the paper is again referring to a concrete (but unstated) iterative algorithm where the model is updated on every timestep, then some planning occurs which impacts the data collected at the next timestep, then repeat. In this case, I see how different planning style leads to learning different models. But all of this is me guessing at what the paper intends.
5. I believe Def'n 1 and Def'n 2 are the same. Probably a typo, but makes it very challenging to formally understand the difference between these. Informally, the distinction is clear (Fig 2 is quite helpful). I love the illustrative example on page 6.
6. Transfer learning appears as an after-thought a few times (sec 4.2.2, 4.3.2, 5.2.2). The layout is counterintuitive. Sec 4 is "Decision-Time vs Background planning" and 4.2 is a clear subsection of that: "Simple instantiations of the two planning styles". But then how is 4.2.2 a subset of 4.2, where 4.2.2 simply defines transfer learning. Is the def'n of transfer learning related to subsection 4.2 or section 4? Perhaps this paragraph instead belongs in Sec 2 or even Sec 1. Relatedly, I see no connection between 4.3.2 and 4.3.
7. Empirical section does not clearly state the hypotheses that it is validating. It refers back to prior sections and implies the hypotheses of study. Would be nice to clearly restate the hypothesis in the empirical section.


I didn't understand the difference between sections 4.2 and 4.3. I see that 4.2 refers to tabular models while 4.3 refers to NN models, however when constructing the analysis toolkit in section 4.1, the paper is very clear that the definitions do not depend on how the model is approximated (i.e. the model class). In this way, shouldn't all later discussion also be agnostic to model class? Theoretically, it seems that the new contribution in section 4.3 is the idea of "improved rollout" policies. That is, we can interpolate between one-step PI and full PI. So to me, Section 4.3 is purely about that interpolation and truly has nothing to do with neural networks. _Perhaps_ this interpolation has historically only been used in papers that also use NN models, but this is merely coincidental to the best of my knowledge.

## Experiments

There are a lot of details about the empirical methodology that are left unstated. Likely many of these details are kept the same as prior works, however the literature is rich and diverse and it is unclear what details are being borrowed from where. This paper should aim to be as complete and explicit as possible. Terms like "training task" and "testing task" are undefined in the standard RL vocabulary. What do they mean here? What inputs does the system receive (i.e. what are the features)? How is state aggregation computed in a two dimensional space?

Section 5.1.1 looks well done. The theory (prop 1) states that background planning should outperform DT planning when $m$ is a PRM. Prop 2 says that if the background planning model is a PXM, and the DT planning model is any arbitrary model, then the background planning should outperform DT planning. Section 5.1.1 sets up conditions in simple gridworld environments that empirically test and validate these claims when the model is learned online and represented tabularly.

I need more details about Sec 5.1.2 to evaluate it. What is a test task and how is it "subsequent"? Does this mean that the episode does not terminate when the agent reaches the first goal state, and it must visit these two goal states in a sequence? Does this mean the environment changed and for the first 25 episodes the goal is in location G and in the remaining episodes the goal is in location G_1? If the env did change, how can either model be a PXM if it contains "stale" data from an old environment? I see some details in App E.1, but these do not clarify what is happening (what is a "training task"?).

Section 5.2.1 seems odd. The hypotheses here are no longer related to the theoretical results from Section 4, but rather are coming from loosely argued conjectures in Section 4. In Section 4.3, it seemed clear that the paper argued "modern instantiation" == "neural network", however Section 5.2.1 purports to study "modern instantiation" with tabular models. This is a case where very clearly stated hypotheses within the empirical section would greatly clarify what is being studied. The conclusions here no longer match with the hypotheses stated in Section 4 (e.g. 4.3.1 with improved rollout), which led me to believe that background planning should always be better. Does this mean the "understanding" goal of the paper failed?

There are many details in the appendix E.2 that I did not understand. For instance, why do we need "bag of words" feature extractors for gridworld environments? What exactly are the inputs to the system, and how do these impact the results?

---

### Decision · Action_Editor_HQ6D · 2024-03-20

**Recommendation:** Reject

**Comment:**

The paper considers the effectiveness of decision-time planning and background planning approaches to model-based reinforcement learning. Decision-time planning uses the environment model to perform search or rollouts to perform planning, whereas background planning employs an estimate of the value function that is continuously updated during model learning based on samples from the model. The paper first provides a theoretical analysis of simple formulations of the two planning approaches, with results that show that background planning outperforms decision-time planning in tabular settings. It then offers hypotheses regarding how these findings will extend to policies that employ neural network-based function approximation. The paper evaluates these results and hypotheses through experiments on both simple domains (MiniGrid) using tabular ("simple") and function approximation-based ("modern") instantiations of planning, and on more challenging domains (Atari and Procgen) using modern (neural network-based) instantiations. The experimental results reveal that decision-time planning does not perform as well as background planning with simple (tabular) instantiations (consistent with the theoretical claims), but that the modern (function approximation-based) instantiations of decision-time planning perform comparable to if not better than background planning.

The paper was evaluated by three reviewers and the reviews resulted in a significant amount of discussion regarding the merits of the paper. The reviewers along the with the AE agree that the goal of providing more insight into the effectiveness of decision-time vs. background planning for model-based RL with tabular and function approximation-based implementations is both admirable and important. Indeed, insight into the conditions under which one approach is superior to the other and why this might be the case is of potential value to practitioners. Notably, this includes the evidence that decision-time planning is superior to background planning when using neural network-based implementations.

However, there are notable weaknesses that persist in the latest version of the paper that call into question the the theoretical and empirical insights into the merits of decision-time vs. background planning. The theoretical analysis relies upon strong assumptions, the validity of which is not fully clear, and claims that should be more rigorously justified. Some reviewers raise the concern about the fact that the theoretical findings are not consistent with the empirical results for the neural network-based implementations (i.e., that decision-time planning performs comparable to or better than background planning), though the AE does not consider this to be a fundamental issue with the paper. As the authors note in their response to the initial set of reviews, a contribution of the paper is in showing under what conditions one might prefer one model-based planning approach over the other. That said, the lack of an in-depth analysis of the differences between neural network-based implementations of the two planning approaches on more modern domains constitutes a missed opportunity to provide important insights into why decision-time planning is superior in these settings. The revisions to the paper along with the authors' response address some of the concerns about the initial submission (notably, through the inclusion of experiments with MuZero and Dreamer on Procgen and Atari domains) and, in that way, help to strengthen the paper, but more work is necessary to solidify the paper's contributions. Specific recommendations include:

* Formalize (mathematically) the core claims and definitions on which the theoretical analysis relies (e.g., that background planning corresponds to an amount of PI that eventually converges to full PI; that function approximators exhibit "good generalization capabilities" in Hypotheses 1, 2, and 3, i.e., what does "good" mean precisely?; in the adaptation setting, how different is the new domain/task?; in Hypothesis 7, "at least a few time steps" is imprecise). While some of the concepts may seem intuitive, making the claims more rigorous would help the reader reason over their validity. This is particularly important for claims/definitions on which the proofs of the propositions depend.
* Related, provide better justification for the assumptions being made (e.g., that the background planning model converges to a perfect model, which makes Proposition 2 seemingly trivial; Theoretical Result 2 follows directly from definitions and assumptions, without a compelling argument as to why those assumptions are fulfilled in settings one cares about; Theoretical Results are not supported by proofs).
* Perform a more controlled set of experiments (e.g., see Reviewer JPaQ's recommendation regarding PlaNet vs. Dreamer) with further analysis to provide more insight into the effectiveness of decision-time vs background planning, particularly when using neural network-based representations.

To summarize, the objectives of the paper in providing insight into the merits of decision-time and background planning approaches to model-based RL are important and of interest to the TMLR community, particularly to practitioners. Improvements that provide a more mathematically rigorous presentation of the theoretical results along with additional experimental analysis are necessary to support the paper's contributions.

**Audience:**

The paper focuses on conditions under which decision-time planning or background planning might be preferred for model-based reinforcement learning, a topic that will be of interest to many in the TMLR community.

**Claims And Evidence:**

The reviewers and AE agree that the goal of providing more insight into the effectiveness of decision-time vs. background planning for model-based RL with tabular and function approximation-based implementations is both admirable and important. The empirical results that show that decision-time planning outperforms background planning in the case of neural network-based implementations is interesting and seems sound. However, as noted below, the paper seems to miss an important opportunity to provide more insight into the performance differences. Meanwhile, the significance of the theoretical results, including the corresponding claims, is unclear due to the assumptions on which the analysis relies.

**Resubmission Of Major Revision:**

The authors may consider submitting a major revision at a later time.